**Investigation**

# Mutant allele formation and inheritance during Cas9/guide RNA-mediated gene drive in a population modification mosquito strain for human malaria control

Rebeca Carballar-Lejarazú,[1,†] Thai Binh Pham,[1,†] Taylor Tushar,[1,†] Anthony A. James [1,2,*]

[1]Department of Microbiology & Molecular Genetics, University of California, Irvine, CA 92697-4025, United States
[2]Department of Molecular Biology & Biochemistry, University of California, Irvine, CA 92697-3900, United States

*Corresponding author: Anthony A. James, Department of Microbiology & Molecular Genetics and Department of Molecular Biology & Biochemistry, 3205 McGaugh Hall, University of California, Irvine, CA 92697-4025, United States. Email: aajames@uci.edu
[†]These authors contributed equally.

Gene-drive systems are under development for the population modification of anopheline vectors of human malaria parasites. The key to their success is the fixation of genes in target mosquito populations that encode molecules preventing parasite transmission. High-efficiency Cas9/guide RNA (gRNA)-based gene-drive systems can facilitate this objective. A potential challenge to these systems is the presence of naturally-occurring or drive system-induced sequence polymorphisms in the genomic target site that could impede Cas9/gRNA-mediated cleavage and negatively impact gene-drive dynamics and fixation. Careful choice of the target site can mitigate the impact of natural variation, and here we analyze drive system-mediated, target-site mutagenesis in the outcross and testcross progeny of an *Anopheles gambiae* strain homo- and hemizygous for the TP13-based gene-drive system. The resulting data allow for estimation of the rates at which drive-system activity generates mutant target-site alleles in the germline and the impact of inherited paternal- and maternal-effect mutations. Functional and nonfunctional mutant alleles are recovered from the germlines at average rates per target gene/generation of 0.08% for paternal and 0.33% for maternal testcross lineages, with an overall average rate of 0.21%. Clustering effects amplify the potential inheritance frequencies of the mutant alleles. Mutations originating in the germlines represent 47% of the total inherited in testcross progeny, with the balance coming from mutant alleles generated by paternal and maternal effects inherited through the respective parental lineages. This approach allows the estimation of potential cleavage-resistant allele formation and inheritance for this drive system in this species and provides empirically derived values to inform more realistic data-driven gene-drive modeling.

Keywords: *Anopheles*; gene-drive systems; resistance alleles; indels; nonhomologous end-joining

## Introduction

One of the potential challenges to the success of CRISPR-based Cas9/guide RNA (gRNA) gene-drive systems for population modification of malaria vector mosquitoes is the presence of target-site polymorphisms that can function as drive-resistant alleles and impact drive dynamics (Gantz et al. 2015; Hammond et al. 2016; Noble et al. 2017; Adolfi et al. 2020; Carballar-Lejarazú et al. 2023; Xu et al. 2025; Yang et al. 2025). These alleles can be natural DNA sequence variants present in the genomes of the targeted species population or can be generated by the drive process during DNA cleavage and repair activity. Some of the single-nucleotide polymorphisms (SNPs) and nucleotide insertion and deletion (indel) mutations in the targeted chromosomal DNA sequence may result in an allele that is resistant to Cas9 nuclease cleavage. Indels are generated when target-site DNA cleavage is repaired by end-joining mechanisms such as nonhomologous end-joining (NHEJ) and microhomology-mediated end-joining (MMEJ), or by recombination following ectopic single-strand annealing (SSA) rather than precise homology-directed repair (HDR; gene conversion) (reviewed in Xue and Greene 2021). If the mutant allele has a lower fitness cost than the drive allele, it could be positively selected and eventually replace the drive system. Therefore, even if high frequencies of drive-carrying alleles are initially achieved, their continued persistence in the population will depend on the presence and fitness of resistance alleles that are pre-existing or generated during the drive process.

We mitigated the impact of naturally-occurring SNPs in the gRNA target site by choosing a DNA sequence that is well-conserved in the *Anopheles gambiae cardinal* gene ortholog (*Agcd*; Carballar-Lejarazú et al. 2020). Furthermore, the natural variants found in a large number (>760) of mosquito samples from geographically-diverse populations were present at low frequencies and were shown to be cleavable in vitro and in vivo by the drive-system components (*Anopheles gambiae 1000 Genomes Consortium et al. 2017*; Carballar-Lejarazú et al. 2020; Tushar et al. 2024). However, cage-trial data show that potential drive-resistant target-site alleles generated during the drive process can impact the ultimate fixation of the drive system in some anopheline species, and release ratios were important in laboratory trials for mitigating these outcomes (Pham et al. 2019; Carballar-Lejarazú et al. 2020, 2023).

Previous attempts to determine the frequencies at which drive-system activity generated potential, drive-resistant mutant

target alleles using an *Anopheles gambiae* strain carrying the core gene-drive system, AgNosCd-1, were confounded by a number of issues including not being able to definitively identify those generated via paternal or maternal effects versus those that occurred in the germline, and by clustering effects that increase the frequency of inheritance of individual mutant alleles (Carballar-Lejarazú et al. 2022). Following up on this work, we performed outcross and testcross experiments with an *An. gambiae* strain carrying the TP13 gene-drive system to attempt to accurately estimate the frequency with which mutations are generated during system activity in the germline. These data also allowed the derivation of the frequency of inheritance of paternal- and maternal-effect mutations. We discuss the findings in the context of mutagenesis and inheritance to highlight the potential impact of drive-resistant allele formation on gene-drive system dynamics.

## Materials and methods
### Mosquito strains and maintenance

The insects used in these experiments include the *An. gambiae* X1 strain (Volohonsky et al. 2015), a mutant strain, *Agcd^{Δ11}* (Carballar-Lejarazú et al. 2022), and the gene-drive carrying line, AgTP13 (Carballar-Lejarazú et al. 2023). AgTP13 has the TP13 autonomous gene-drive system comprising the AgNosCd-1 gene-drive components that include a U6-expressed gRNA targeting the *An. gambiae cardinal* gene ortholog, *Agcd*, the Cas9 nuclease driven by the *An. gambiae nanos* gene promoter, which mediates germline expression, and the cyan fluorescent protein (CFP) gene driven by the 3xP3 synthetic promoter as a dominant phenotypic marker (Horn and Wimmer 2000; Carballar-Lejarazú et al. 2020; Terradas et al. 2022). These are linked to a pair of engineered genes encoding the mosquito codon bias-optimized, murine single-chain antibodies, m1C3 and m2A10, targeting *Plasmodium falciparum* ookinetes and sporozoites, respectively (Isaacs et al. 2011). The *Agcd^{Δ11}* mutant strain is derived from the *An. gambiae* G3 strain and has an 11-basepair (bp) deletion in the *Agcd* coding region that confers resistance to Cas9 cleavage; mosquitoes homozygous for this mutation have a *cardinal*, red-eye phenotype (Carballar-Lejarazú et al. 2020, 2022). Mosquitoes were maintained in an insectary at 27 °C with 75% humidity and a 12-h day/night with a 1-h dusk/dawn lighting cycle. Adult females were provided with rabbit blood (Colorado Serum Company, CO, USA) through artificial membranes (Hemotek, Inc., Blackburn, UK).

### Nomenclature

The visible phenotypes monitored in these analyses include the larval, pupal, and adult "black" eyes that are conferred by a dominant, wild-type *Agcd* allele, *cd^+* (Supplementary Fig. 1; Carballar-Lejarazú et al. 2022). Recessive nonfunctional mutant alleles, designated *cd^-*, confer a red-eye phenotype when homozygous, which is easily visible in larvae, pupae, and recently emerged adults. The adult eye darkens over a 2-d period following emergence to a near wild-type phenotype due to the nonenzymatic oxidation of 3-hydroxykynurenine to xanthommatin in the branch of the tryptophan-metabolic pathway that leads to the production of eye pigments (Li et al. 1999). An additional phenotype, designated CFP^+, is conferred by the expression of the dominant marker gene encoding the CFP that indicates the presence of the drive allele and is easily detected in all life stages, and its absence is designated CFP^-. The final phenotype, a mosaic of red and black ommatidia visible in pupal and adult eyes, is designated "tear" eye and results from paternal and maternal effects caused by vertical transmission of Cas9/gRNA complexes in sperm and eggs, respectively, that generate clones of mutant alleles of variable size in wild-type backgrounds in single mosquitoes (Fig. 1; Carballar-Lejarazú et al. 2020, 2022, 2023). "Expected" phenotypes are those whose frequencies and/or presentation are predicted by canonical Mendelian segregation and inheritance, whereas "exceptional" refers to those that are not (e.g. tear/mosaic phenotypes).

The phenotypic designations alone are not sufficient to categorize the complexity of the corresponding alternative genotypes. Therefore, 4 overlapping molecular genotypes were adapted in part from Sánchez C et al. (2020). The presence of the drive allele is designated here as "D" ("H" in the Sánchez description), which is phenotypically positive for the dominant CFP^+ marker gene and confers a recessive red-eye phenotype in drive-system homozygotes (D/D) as a result of the drive-system insertion in *Agcd* that disrupts the open-reading frame (ORF). CFP^+ mosquitoes with red eyes can also be produced in heteroallelic combinations with *cd^-* alleles. An intact wild-type allele, W, produces the previously described dominant black-eye phenotype. In addition, a dominant black-eye phenotype is conferred by a functional mutant allele, designated "R," resulting typically from small in-frame indels at the *Agcd* gRNA target site that preserve the ORF (Carballar-Lejarazú et al. 2020, 2023). Finally, a nonfunctional, "broken" allele, "B," confers a recessive red-eye phenotype when homozygous, opposite a D allele or in a heteroallelic combination with another B allele. B alleles result typically from out-of-frame indels at the gRNA target site that disrupt the ORF or from large in-frame indels (Carballar-Lejarazú et al. 2022, 2023).

"Target" allele refers to any wild-type allele, W/*cd^+*, in a mosquito in circumstances where it could be subject to Cas/9/gRNA-mediated cleavage that induces DNA repair. The inheritance dynamics of the gene-drive system are expressed as the gene-drive inheritance (GDI), which is measured as the percentage of CFP^+ progeny of the total in any cross where 1 or both parents carry 1 or 2 copies of the gene-drive system. In addition, mosquitoes hemizygous for the gene-drive system (CFP^+/*cd^+*; D/W) outcrossed to wild-type mosquitoes (CFP^-/*cd^+*; W/W) allow the calculation of the percentage of progeny inheriting an HDR-mediated "converted" gene-drive target allele, cGDI [cGDI = (GDI−50)/50 × 100], which is the percentage of progeny carrying the gene-drive system in excess of what would be expected by Mendelian segregation of the alternative alleles (CFP^+/*cd^+*; D/W) in the parents and serves as an index of gene-drive efficiency ("cGDI" is also referred to as "homing," e.g. Hammond et al. 2016).

### Mosquito mating and screening
#### Homozygote outcrosses

A series of independent outcross replicates were made between homozygous AgTP13 mosquitoes and wild-type mosquitoes (Fig. 1). Mating replicates consisted of crosses of 50 gene-drive males or females to 50 *An. gambiae* X1 of the opposite sex to establish separate male and female founder lineages. All mosquitoes were 3 to 5 d posteclosion at the time of mating, were allowed to mate for a total of 3 d, and females were subsequently given 2 consecutive blood meals over a period of 2 d. Females were allowed 3 d to digest the blood meal, after which they were provided with an egg cup and allowed to oviposit for 3 d. Emergent first-instar larvae were placed in plastic containers ("pans," 11″ [7.94 cm] × 6″ [15.24 cm] × 4″ [10.16 cm]) containing 1 L of distilled water at a

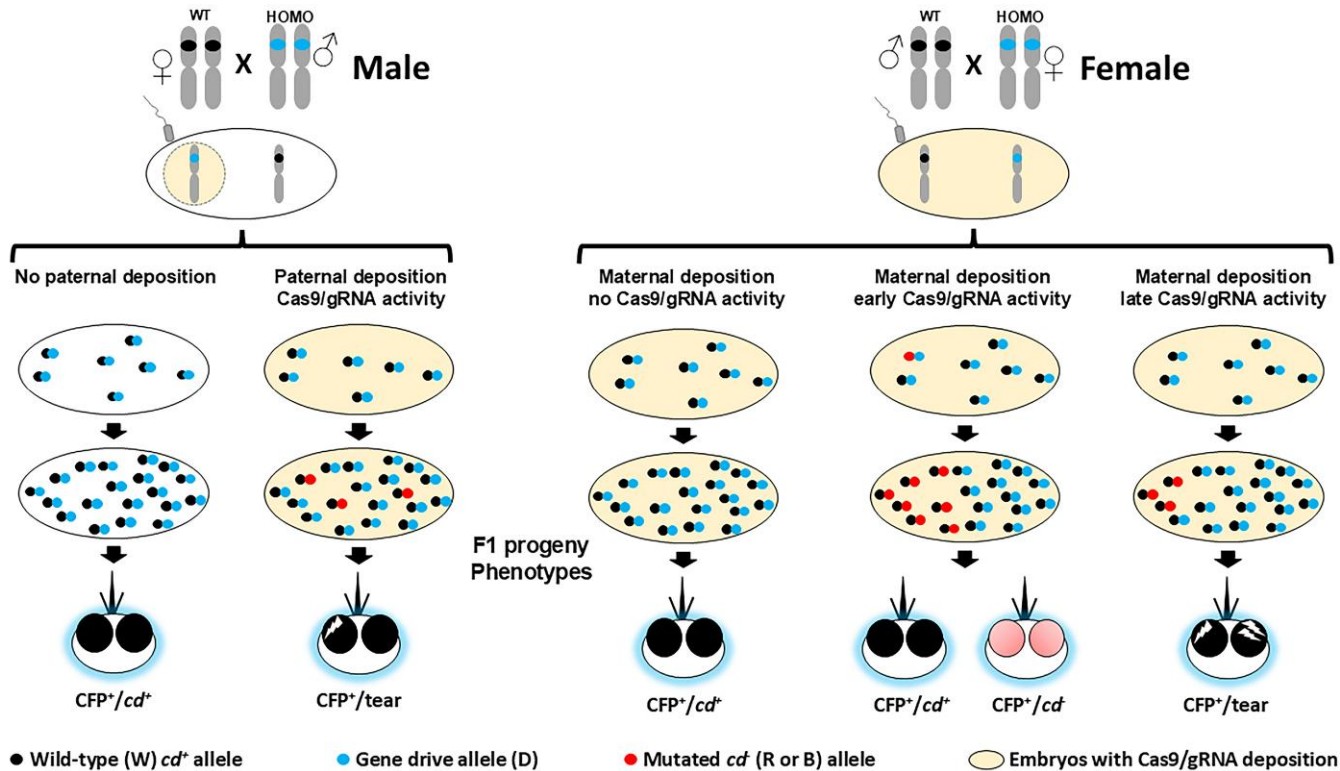

**Fig. 1.** Schematic representations of the crosses and phenotypic outcomes of male- and female-lineage gene-drive outcrosses. Male lineage (left). *Top*: homozygous gene-drive carrying adult male mosquitoes are outcrossed to wild-type females. The vertical elongated double ellipses (gray) represent the *An. gambaie* third chromosome, the small black ovals on the female (♀) chromosomes represent a wild-type *cardinal* gene allele and the blue ovals on the males (♂) represent the integrated TP13 gene-drive system in the *cardinal* gene. The large horizontal white oval represents a recently fertilized, hemizygous embryo carrying the female-contributed third chromosome with a wild-type *cardinal* allele and a male pronucleus (light-tan circle) with a drive-carrying third chromosome (blue oval) and potentially carrying Cas9/gRNA complexes. *Middle*: diploid nuclei in the fertilized embryos undergo 10 to 12 rounds of mitosis before migrating to the cortex to form the blastoderm. The compound black/blue circles represent diploid preblastoderm hemizygous nuclei in embryos without (white) or with (light-tan) paternally donated Cas9/gRNA complexes. At any time during this early development, some genes in embryos with Cas9/gRNA complexes could be mutated to nonfunctional, B, alleles (small red circles). R alleles also can be formed, but are not distinguishable phenotypically from the wild-type alleles in the later stages of development. *Bottom*: The resulting adults all have a drive allele conferring blue fluorescent eyes (CFP[+], blue halo), with the majority also having wild-type-colored eyes (*cd[+]*, black circles). A small percentage of those coming from embryos with paternally deposited Cas9/gRNA complexes have functional (R) or nonfunctional (B) mutant alleles (red circles). The B mutant alleles can be included in clones of cells that form the eyes, leading to the mosaic, tear-eye phenotype (black circle with white slash). Female lineage (right). *Top*: Homozygous female gene-drive carrying adult mosquitoes are outcrossed to wild-type males. All schematic representations are the same as the male lineage, with the additions that the large horizontal light-tan oval represents a recently fertilized, hemizygous embryo with maternally deposited Cas9/gRNA complexes that carries the female-contributed drive-carrying third chromosome and a paternally donated chromosome with a wild-type *cardinal* gene allele. *Middle*: At any time during embryonic development, some wild-type alleles could be mutated to R or B alleles. *Bottom middle*: If DNA cleavage and repair happens early, large clones of cells that contribute to the adult eye result in a black eye if the mutation results in a functional R allele or a red-eye mosquito (large pink circles) if the mutation produces a nonfunctional B allele. *Bottom right*: Late cleavage and repair events may generate smaller clones that contribute a portion of cells that make up the adult eye and result in a mosaic, tear-eye phenotype. CFP[+], cyan fluorescent protein positive; *cd[-]*, mutant *cardinal* gene; *cd[+]*, wild-type *cardinal* gene; D, drive allele; R, functional mutant allele; B, nonfunctional mutant allele; W, wild-type allele.

density of ~200 larvae per pan. All mating experiments were performed in 3 independent replicates unless indicated otherwise, and the total progeny embryos generated from each mating, designated F1, were hatched and reared to the pupal stage. Individual pupae were removed daily from the pans, screened phenotypically using light (Leica MZ6) and fluorescent (Leica M165 FC) microscopy, and sorted by eye-color phenotype (red eye [*cd[-]*], black eye [wild-type, *cd[+]*], or tear [mosaic]). The CFP marker (positive, CFP[+], or negative, CFP[-]) was used as an indicator of the presence or absence, respectively, of the drive system. The sorted pupae were segregated by sex and placed separately in 16 oz (~473 mL) cups in 0.216 m³ disposable cages and allowed to emerge over 2 d (Pham et al. 2019). Virgin adults of each scored phenotype were used either in subsequent mating experiments and then preserved or preserved immediately at −20 °C for molecular analyses.

### Hemizygote testcrosses

Multiple reciprocal replicate testcrosses were conducted when possible by crossing segregated F1 phenotypic groups of 50 AgTP13 hemizygous gene-drive males or females to 50 homozygous *Agcd[411]* mosquitoes of the opposite sex (Fig. 2). The small numbers of recovered F1 CFP[+]/tear (mosaic) mosquitoes from the male lineages allowed only 10 to 15 F1 males or females to be mated in a single cross to 50 *Agcd[411]* members of the opposite sex. Similarly, only 20 to 40 F1 CFP[+]/*cd[-]* (red-eye) males or females from female lineages were mated to 50 *Agcd[411]* members of the opposite sex for each cross. Rearing procedures were identical to those of the outcrosses. All progeny, designated F2, were scored for phenotypes, and representative samples of expected and exceptional progeny were saved for molecular genotyping.

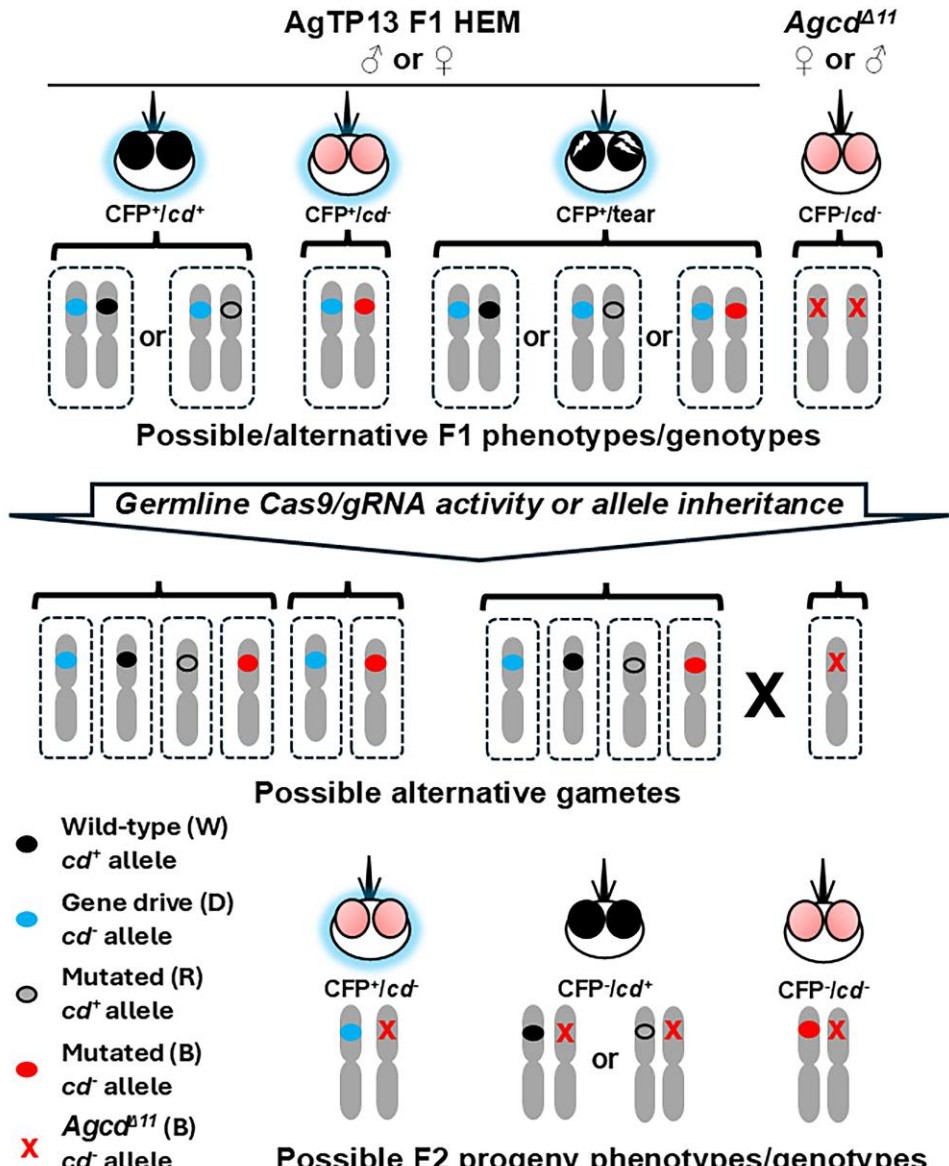

**Fig. 2.** Schematic representations of the crosses and phenotypic outcomes of male- and female-lineage gene-drive testcrosses. *Top*: Possible alternative AgTP13 hemizygous gene-drive F1 (AgTP13 F1 HEM) and homozygous *Agcd^Δ11* (*Agcd^Δ11*) parental phenotypes and genotypes mated in the testcrosses. Images and colors are identical to those used in Fig. 1, with the addition of an empty small oval on the third chromosome representing a known, functional R allele and a red "X" representing the *Agcd^Δ11* allele. Blue halos indicate CFP+ genotypes/phenotypes, and no halos are CFP⁻. *Middle*: germline activity of the Cas9/gRNA complexes can result in gametes carrying target alleles converted to gene-drive, NHEJ, MMEJ, or alt-MMEJ functional, R, or nonfunctional, B, mutant alleles. Alternatively, the germline could carry an R or B allele inherited from the F1 parent. *Bottom*: these events can be detected in the F2 progeny of the testcrosses. CFP+, cyan fluorescent protein positive; *cd⁻*, mutant *cardinal* gene; *cd+*, wild-type *cardinal* allele or functional mutant, R, allele; D, drive allele; B, nonfunctional mutant allele; NHEJ, nonhomologous end-joining; MMEJ, microhomology-mediated end-joining; alt-MMEJ, alternative MMEJ; W, wild-type allele.

## Molecular genotypic analysis of individual mosquitoes

The detection protocol for identifying drive system-mediated mutant alleles was adapted from Carballar-Lejarazú et al. (2022). Genomic DNA was extracted from individual mosquitoes using an Extract-N-Amp PCR kit (Sigma). Samples were homogenized in 60 µL of a solution with a 4:1 ratio of Extract:tissue preparation. The mixture was incubated for 20 min at 55 °C followed by 10 min at 95 °C. Sixty microliters of Neutralization B solution were added to the mixture, followed by vortexing for 20 s and 10 min of centrifugation. Two microliters of the extracted DNA were used for each gene amplification (PCR) reaction in a total volume of 30 µL. PCR was performed with the DreamTaq PCR mix (Thermo-Fisher) according to the manufacturer's protocol. A fragment spanning the *Agcd* target site was amplified with the oligonucleotide primers 5′GTACTCGTACGGTCGCTCCTTA3′ and 5′ ATTGTTGTTGCAGATGAGTCGT3′ to determine if the target site was wild type or mutated. Another fragment spanning the junction between the left *Agcd* homology arm and the drive cassette was also amplified (primers 5′CTCCAGCAGCTCCTTATCGC3′ and 5′ GCTTGTTTGAATTGAATTGTCGC3′) to confirm the presence or absence of the drive allele (Carballar-Lejarazú al. 2023). Amplicons were subjected to Sanger sequencing (Azenta, USA), and the results were analyzed for potential SNPs and indels at the gRNA target site

(Snapgene and Indigo, https://www.gear-genomics.com/indigo/, Rausch et al. 2020).

## Statistical analysis

Chi-square tests were used where appropriate to determine significance.

## Results

### Inheritance dynamics and phenotypic analyses of F1 progeny of AgTP13 outcrosses

Outcrosses of homozygous gene-drive-carrying AgTP13 males or females to wild-type members of the opposite sex from the *An. gambiae* X1 strain were performed to generate hemizygous parental mosquitoes required for subsequent testcrosses (Fig. 1). Separate male and female founder lineages were set up to monitor possible sex-specific effects. The progeny of these crosses were analyzed both phenotypically and genotypically to account for mutant target-site alleles that could be transmitted to the progeny of the subsequent testcrosses.

A total of 5,251 F1 progeny, 3,092 from the male founder and 2,159 from the female founder lineages, were screened for the presence of the gene-drive system and their eye color phenotypes (Table 1). The GDI values, representing the overall inheritance of the gene-drive system, were 100% as expected for both homozygous male and female founder outcrosses. The small percentage, 0.9% (28/3,092), of exceptional F1 progeny from the male founder and the much larger percentage, 53.3% (1,151/2,159), from the female founder outcrosses with tear (mosaic) phenotypes, are consistent with the respective paternal and maternal deposition of stable Cas9/gRNA complexes through the gametes (sperm and eggs) that acted on target chromosomes in the resulting embryos (Fig. 1; Carballar-Lejarazú et al. 2020, 2022, 2023). Furthermore, unlike the male founder crosses, F1 CFP$^+$/$cd^-$ progeny were recovered from the female outcrosses at a frequency of 3.1% (67/2,159; Table 1). These progeny are consistent with the conclusion that maternally deposited Cas9/gRNA complexes in the oocytes acted early on the incoming, male-donated wild-type target alleles in the resulting embryos to produce 1 or more large $cd^-$ (B) clones that encompassed completely the adult eye, and opposite a CFP$^+$ drive allele, conferred the mutant, $cd^-$, red-eye phenotype (Fig. 1; Carballar-Lejarazú et al. 2022).

### Genotypic analyses of F1 progeny of AgTP13 outcrosses

Genotypic analyses were carried out on select F1 progeny to identify mutant alleles that could be transmitted vertically to the progeny of the testcrosses and complicate the estimates of the formation de novo of mutations in the F1 germline cells. Therefore, following the phenotype-based matings in the testcrosses described below, samples of F1 mosquitoes from the male and female founder outcrosses with expected and exceptional phenotypes were sequenced. The genotypes of 587 of these, 467 with expected phenotypes and 120 with exceptional phenotypes, were determined (Table 2). As expected, all 258 (100%) CFP$^+$/$cd^+$ mosquitoes sequenced from the male founder lineage carried drive (D) and wild-type (W) alleles, as did 200/209 (95.7%) of the CFP$^+$/$cd^+$ samples from the female founder outcross. The remaining 9 CFP$^+$/$cd^+$ female-lineage progeny had R alleles (described below). Sequences from 26 of the male-lineage F1 CFP$^+$/tear progeny also had a distinct W allele; however, the genotypes are designated "cryptic" to indicate the clear presence of a wild-type allele in the sequencing trace and the high probability of the presence of a B allele refractory to sequencing that accounts for the eye being scored as a tear (Supplementary Fig. 2). Similarly, a large percentage, 54.5% (36/66), of the female-lineage F1 CFP$^+$/tear progeny were cryptic and an additional 39% (26/66) were designated "disturbed," a label that refers to sequencing traces with multiple, unresolvable mutant alleles that must include combinations of B with R or W alleles to produce the tear phenotype (Table 2).

### *General sequence properties of mutant target-site alleles*

No discernible mutant genotypes were recovered from male-lineage F1 progeny, but 41 samples with CFP$^+$/$cd^+$, CFP$^+$/tear or CFP$^+$/$cd^-$ phenotypes from female-lineage F1 CFP$^+$ progeny were sequenced (Supplementary Table 2; sequences are labeled alphanumerically, "OC-xxx," to indicate that they are from the outcrosses; CFP$^+$ and W allele sequences are not listed). Fourteen of the 41 samples were genotypically mosaic for either 2 mutant alleles or 1 mutant allele and 1 wild-type allele, along with the expected CFP$^+$ allele. In total, 55 alleles were identified, 2 of which were W, and the remaining 53 were R or B mutations. Eight alleles were detected in multiple samples from different crosses and replicates and are designated a to h in Supplementary Table 2.

Detailed examination of the mutant sequences revealed that 20 of them most likely resulted from NHEJ repair events, 7 of which were indels. Nineteen sequences were deletions that had characteristics consistent with MMEJ (deleted sequences flanked by short, ≥2 bp, direct DNA repeats in the wild-type sequence). The remaining 14 mutations, 2 R (OC-1091/1423) and 12 B (OC-1139/1140/1141/1144/1147/1148/1154/1159/1165/1167/1168/1169), were indels with sequences of ≥3 bp that were direct or inverted repeats of nearby genomic DNA. These mutations have been interpreted to result from 1 of 3 possible alternative outcomes of MMEJ (designated here as "alt-MMEJ") during which DNA synthesis following annealing of the microhomologous regions can result in duplications of adjacent DNA sequences (Ceccaldi et al. 2016).

A total of 35 of the 53 identified mutant alleles had an intact protospacer adjacent motif (PAM) site sequence (designated PAM$^+$), which is necessary for Cas9/gRNA cleavage (Supplementary Table 2; Jiang and Doudna 2017). Nine of these, OC-929/1066/1091/1156/1162/1163/1166/1172/1411, had in addition conserved the 3 nucleotides proximal to 3′-end of the PAM site. It has been proposed that this class of alleles, PEPPR

**Table 1.** F1 progeny phenotypes and inheritance rates in AgTP13×wild-type outcrosses[a,b].

| Parental cross | CFP$^+$/$cd^+$ | CFP$^+$/$cd^-$ | CFP$^+$/tear | CFP$^-$/$cd^+$ | CFP$^-$/$cd^-$ | Tot | GDI% |
|---|---|---|---|---|---|---|---|
| AgTP13 homozygous (CFP$^+$/$cd^-$; D/D) ♂×WT-X1 (CFP$^-$/$cd^+$; W/W) ♀ | 3,064 (99.1%) | 0 | 28 (0.9%) | 0 | 0 | 3,092 | 100 |
| AgTP13 homozygous (CFP$^+$/$cd^-$; D/D) ♀×WT-X1 (CFP$^-$/$cd^+$; W/W) ♂ | 941 (43.6%) | 67 (3.1%) | 1,151 (53.3%) | 0 | 0 | 2,159 | 100 |

CFP$^+$, cyan fluorescent protein positive; $cd^+$, black eye, wild-type allele of *cardinal*; $cd^-$, red eye, nonfunctional mutant allele of *cardinal*; GDI, gene-drive inheritance frequency.
[a]Combined results from 3 replicates for each outcross.
[b]Results from individual replicates are listed in Supplementary Table 1.

**Table 2.** Summary of molecular genotyping of F1 progeny of homozygous AgTP13 outcrosses.

| Parental cross | Progeny phenotype | No. sequenced[b] | Genotypes[a] | | | | | | | | |
|---|---|---|---|---|---|---|---|---|---|---|---|
| | | | D/W | D/R | D/B | D/W/R | D/W/B | D/R/B | D/B/B | cryptic (D/W/B*) | disturbed (D/B*) |
| **F1 progeny from the AgTP13 male outcross** AgTP13 homozygous (CFP+/cd+; D/D) ♂ × WT-X1 (CFP−/cd+; W/W) ♀ | CFP+/cd+ | 258 | 258/258 (100%) | | | | | | | | |
| | CFP+/tear | 26 | | | | | | | | 26/26 (100%) | |
| **F1 progeny from the AgTP13 female outcross** AgTP13 homozygous (CFP+/cd+; D/D) ♀ × WT-X1 (CFP−/cd+; W/W) ♂ | CFP+/cd+ | 209 | 200/209 (95.7%) | 8/209 (3.8%) | | | | 1/209 (0.5%) | | | |
| | CFP+/tear | 66 | | | | 1/66 (1.5%) | 1/66 (1.5%) | 2/66 (3%) | | 36/66 (54.5%) | 26/66 (39.3%) |
| | CFP+/cd− | 28 | | | 19/28 (67.8%) | | | 1/28 (3.6%) | 8/28 (28.6%) | | |

CFP+, cyan fluorescent protein positive; cd+, black eye, wild-type cardinal allele; cd−, red eye, nonfunctional mutant cardinal allele; D, drive allele; W, wild-type allele; R, functional mutant allele; B, nonfunctional mutant allele; B*, likely nonfunctional mutant allele based on the phenotype. Progeny with more than 2 alleles with "cryptic" and "disturbed" traces are phenotypically tear mosaics.
[a]Number and percent with genotype.
[b]Number of samples collected and sequenced from all replicates of each cross.

(Pam-End Proximal Protected Repair), results from prolonged binding of the Cas9/gRNA complex to the target site after cleavage that protects the exposed DNA strands from exonuclease degradation (discussed in Li et al. 2024). The design of the crosses with uncleavable D alleles in the homozygous drive-carrying female parents supports the conclusion that all 53 of the listed mutant alleles, 13 R and 40 B, arose from independent Cas9/gRNA-mediated maternal-effect events on target alleles in the embryo. Furthermore, paternal and maternal effects are likely responsible for the cryptic and disturbed genotypes recovered from the respective male and female lineages (Table 2).

### Specific properties of R target allele mutations

Ten of the 13 R alleles sequenced in the female-lineage F1 progeny restore the ORF with deletions of 3, 6, 9, 12, or 15 bp (Supplementary Table 2). Three samples (OC-1066 [−5, +2 bp], OC-1091 [−7, +7 bp], and OC-1423 [−11, +5 bp]) have compensatory indels that also restore the ORF. Four R allele sequences, OC-1084/1348 (−6 bp), OC-929/1411 (−6 bp), OC-945/1369 (−9 bp), and OC-1154/1392 (−12 bp), were seen in 2 samples each from different replicates or crosses. The OC-945 sample is genotypically mosaic for a 9-bp deletion R allele accompanied by a 14-bp deletion B allele; however, it was not scored phenotypically as tear because the B clone did not encompass the eye (Table 2). All sequences present in multiple samples result from putative MMEJ except the OC-929/1411 pair.

### Specific properties of B target allele mutations

All 40 of the sequenced B alleles in the female-lineage F1 progeny encode nonfunctional proteins resulting from disruptions of the ORF by indels, introductions of stop codons (OC-1159/1153), or large, ≥21 bp, in-frame deletions (OC-1146/1161 [−21 bp], OC-1149 [−24 bp], and OC-1159 [−36 bp]) (Supplementary Table 2). Three alleles, OC-1152/1170 (−11 bp), OC-1146/1161 (−21 bp), and OC-1121/1160 (−29 bp), were detected in 2 different samples, and a fourth, OC-1066/1162/1163/1172 (−13 bp), was seen in 4 samples. All B sequences present in multiple samples have characteristics of MMEJ events. The 3 MMEJ alleles with deletions of 11 (OC-1152/1170), 13 (OC-1066/1162/1163/1172), and 14 (OC-945) bp were seen previously in other experiments performed in our laboratory, and the 11-bp deletion $Agcd^{411}$ strain was recovered and established from this previous work (Supplementary Table 3).

Some samples have complex combinations of mutant alleles. For example, OC-1066 was scored as tear and has a 13-bp deletion B allele along with the previously noted R allele comprising a 5-bp deletion and 2-bp insertion that restores the ORF to produce the mosaic phenotype. OC-1091 was also scored as tear and has a 31-bp deletion B allele accompanied by the previously noted R allele with a 7-bp deletion and 7-bp insertion. Sample OC-1154 has a 12-bp deletion R allele and a −1, +9 B indel and was scored as CFP+/cd−, presumably due to the R clone not encompassing the eye. OC-1157 was also scored with the CFP+/cd− phenotype but has what appears to be an R allele comprising a 12-bp deletion accompanied by a 3-bp insertion, thus presumably restoring the ORF. This phenotypic/genotypic combination is consistent with the conclusion that not all mutations that preserve the ORF result in genes whose products are functionally capable of restoring the wild-type eye phenotype.

### Inheritance of F1 mutant target alleles

Twenty-one of the combined R and B mutant alleles sequenced are seen in the subsequent F2 testcross progeny (highlighted in

**Table 3.** Percentages of F2 progeny phenotypes in AgTP13×*Agcd^A11* testcrosses[a,b].

| Parental cross | CFP⁺/*cd*⁺ | CFP⁺/*cd*⁻ | CFP⁺/tear | CFP⁻/*cd*⁺ | CFP⁻/*cd*⁻ | Tot | GDI/cGDI% |
|---|---|---|---|---|---|---|---|
| **Male lineage** | | | | | | | |
| F1 CFP⁺/*cd*⁺ ♂×*Agcd^A11* ♀ | 0 | 3,486 (99.8%) | 0 | 4 (0.11%) | 1 (0.028%) | 3,491 | 99.8/99.7 |
| F1 CFP⁺/*cd*⁺ ♀×*Agcd^A11* ♂ | 0 | 2,930 (96.8%) | 0 | 91 (3%) | 5 (0.16%) | 3,026 | 96.8/93.6 |
| F1 CFP⁺/tear ♂×*Agcd^A11* ♀ | 0 | 230 (98.7%) | 0 | 0 | 3 (1.3%) | 2,33 | 98.7/97.4 |
| F1 CFP⁺/tear ♀×*Agcd^A11* ♂ | 0 | 384 (98.9%) | 0 | 3 (0.77%) | 1 (0.26%) | 388 | 98.9/97.9 |
| **Female lineage** | | | | | | | |
| F1 CFP⁺/*cd*⁺ ♂×*Agcd^A11* ♀ | 0 | 1,790 (94.1%) | 0 | 81 (4.3%) | 30 (1.6%) | 1,901 | 94.1/88.3 |
| F1 CFP⁺/*cd*⁺ ♀×*Agcd^A11* ♂ | 0 | 1,886 (91.3%) | 0 | 139 (6.7%) | 39 (1.9%) | 2,064 | 91.3/82.7 |
| F1 CFP⁺/tear ♂×*Agcd^A11* ♀ | 0 | 1,620 (98%) | 0 | 6 (0.36%) | 27 (1.6%) | 1,653 | 98/96 |
| F1 CFP⁺/tear ♀×*Agcd^A11* ♂ | 0 | 2,011 (96.1%) | 0 | 58 (2.8%) | 24 (1.1%) | 2,093 | 96.1/92.1 |
| F1 CFP⁺/*cd*⁻ ♂×*Agcd^A11* ♀ | 0 | 167 (50.6%) | 0 | 0 | 163 (49.4%) | 330 | 50.6/NA |
| F1 CFP⁺/*cd*⁻ ♀×*Agcd^A11* ♂ | 0 | 276 (52.7%) | 0 | 29 (5.5%) | 219 (41.8%) | 524 | 52.7/NA |

NA, not applicable; CFP⁺, cyan fluorescent protein positive; *cd*⁺, black eye, wild-type allele of *cardinal*; *cd*⁻, red eye, nonfunctional mutant allele of *cardinal*; GDI, gene-drive inheritance frequency; cGDI, converted gene-drive inheritance.
[a]Combined results from 3 replicates for each testcross.
[b]Results from individual replicates are listed in Supplementary Table 4.

green in Supplementary Table 2). Sequences of 7 of these, OC-945/978/1141/1156/1157/1159/1169, are seen only once in the F1 generation progeny. The remaining 14 sequences represent 2 each in 7 of the 8 multiple groups identified (a to e, g, and h; the f group was seen only in the outcross progeny; Supplementary Table 2). Sequences of 1 allele each from the 5 mosaic genotypes of OC-1146/1154/1159/1169/1170 and both alleles in OC-945 were seen in the next generation. These data are important for estimating independent germline mutation events in the following testcrosses.

## Inheritance dynamics and phenotypic analyses of F2 progeny of hemizygous AgTP13 testcrosses

Following the outcrosses, hemizygous CFP⁺ F1 AgTP13 adult progeny from all phenotypic classes, wild-type (*cd*⁺), tear (mosaic), or red eye (*cd*⁻), of the male and female founder lineages were crossed separately to members of the opposite sex of the homozygous *Agcd^A11* mutant strain (Fig. 2). As noted, *Agcd^A11* is a drive-resistant B allele that is uncleavable by paternally or maternally inherited Cas9/gRNA complexes (Carballar-Lejarazú et al. 2020, 2022), therefore all potential target alleles in the testcrosses come from the germlines of the hemizygous drive-carrying F1 parents. A total of 15,703 F2 progeny, 7,138 from the male founder lineage and 8,565 from the female founder lineage, were screened for the inheritance of the gene-drive system and eye-color phenotypes, and exceptional progeny were saved for molecular genotyping (Fig. 2; Table 3). These data allow the calculation of germline mutation rates following adjustments to account for inherited mutant alleles and clustering effects.

### General phenotypic properties of male founder lineage F2 testcross progeny

Male-lineage, male-parental CFP⁺/*cd*⁺ testcrosses had a GDI of 99.8% and a cGDI of 99.7%, with the latter being consistent with high-efficiency, gene-drive-mediated HDR gene conversion in the germline and small percentages of target alleles not being converted to CFP⁺ (CFP⁻/*cd*⁺ = 0.11%; CFP⁻/*cd*⁻ = 0.028%) (Fig. 2; Table 3). Male-lineage, female-parental CFP⁺/*cd*⁺ testcrosses had GDI (96.8%) and cGDI (93.6%) values slightly lower than the males, and this sex-dependent difference was observed previously (Carballar-Lejarazú et al. 2020, 2023). Similar to the male testcrosses, small percentages of target alleles were not converted to CFP⁺ (CFP⁻/*cd*⁺ = 3.0%; CFP⁻/*cd*⁻ = 0.16%).

The low frequency CFP⁻ phenotypes in the male parental crosses could arise from nonconversion of the target allele (*cd*⁺), mutant R or

B alleles (*cd*⁺ and *cd*⁻) resulting from paternal deposition of Cas9/gRNA complexes into an oocyte via a spermatozoon carrying a wild-type allele and subsequent DNA cleavage and repair in the resulting embryo, or the presence of genotypically mosaic F1 germlines. Similarly, the *cd*⁺ alleles in the progeny of the female parental crosses could result from nonconversion of the target alleles or be mutant R alleles resulting from maternal deposition of Cas9/gRNA complexes into oocytes and later DNA cleavage and repair; maternal deposition also could explain the *cd*⁻ (B) alleles. Here again, there is the possibility of genotypically mosaic F1 germlines.

Male-lineage, male and female CFP⁺/tear testcrosses had respective GDI/cGDI percentages of 98.7%/97.4 and 98.9%/97.9%, and the reductions in the male parental values compared to the CFP⁺/*cd*⁺ crosses likely result from cleavage-resistant R or B alleles in the germlines (Fig. 2; Table 3). The small percentage, 0.77%, of CFP⁻/*cd*⁺ wild-type phenotypes recovered in the F2 progeny from female tear testcross was validated later as genotypic wild types (W; Table 4). Small percentages, 1.3% and 0.26%, of CFP⁻/*cd*⁻ (red-eye) mosquitoes were recovered from male and female tear testcrosses, respectively, and as described for the CFP⁺/*cd*⁺ testcrosses, there are a number of ways that these could have arisen, with the most likely explanation being genotypic mosaicism in the parental F1 germlines. No phenotypically tear mosaic F2 progeny were detected in any of the crosses, which is expected because the drive-resistant *Agcd^A11* allele is uncleavable.

### General phenotypic properties of female founder lineage F2 testcross progeny

Crosses from the F1 female founder lineages had GDIs/cGDIs of 94.1%/88.3% (CFP⁺/*cd*⁺ male parent) and 91.3%/82.7% (CFP⁺/*cd*⁺ female parent) with CFP⁺/tear crosses having GDIs/cGDIs of 98%/96% and 96.1%/92.1% for male and female parents, respectively (Fig. 2; Table 3). As with the male founder lineage crosses, no tear mosaic F2 progeny were detected as a consequence of the uncleavable drive-resistant *Agcd^A11* mutation. Wild-type (CFP⁻/*cd*⁺) phenotypes were recovered from both male, 4.3% (81/1,901), and female, 6.7% (139/2,064), CFP⁺ wild-type (*cd*⁺) testcrosses. This phenotype was also recovered from the CFP⁺/tear male, 0.36% (6/1,653), and female, 2.8% (58/2,093), testcrosses. Notably, these percentages are 2- to 39-fold higher than what was found in the male founder lineages and result most likely from the previously described maternal effects that generated cleavage-resistant R alleles present in the hemizygous drive parent. Mosquitoes with a CFP⁻/*cd*⁻, (red-eye) phenotype also were recovered in these testcrosses (CFP⁺/*cd*⁺ male parent: 1.6% [30/1,901]; CFP⁺/*cd*⁺ female

**Table 4.** Summary of molecular genotyping of F2 progeny of AgTP13×$Agcd^{A11}$ testcrosses.

| Parental cross | Progeny phenotype | No. sequenced[b] | Genotypes[a] | | | | |
|---|---|---|---|---|---|---|---|
| | | | D/$cd^{A11}$ | W/$cd^{A11}$ | R/$cd^{A11}$ | B/$cd^{A11}$ | B/B/$cd^{A11}$ |
| **F1 male lineage** | | | | | | | |
| F1 CFP⁺/$cd^+$ ♂×$Agcd^{A11}$ ♀ | CFP⁺/$cd^-$ | 60 | 60/60 (100%) | | | | |
| | CFP⁻/$cd^+$ | 3 | | 2/3 (67%) | 1/3 (33%) | | |
| | CFP⁻/$cd^-$ | 1 | | | | 1/1 (100%) | |
| F1 CFP⁺/$cd^+$ ♀×$Agcd^{A11}$ ♂ | CFP⁺/$cd^-$ | 60 | 60/60 (100%) | | | | |
| | CFP⁻/$cd^+$ | 82 | | 82/82 (100%) | | | |
| | CFP⁻/$cd^-$ | 5 | | | | 5/5 (100%) | |
| F1 CFP⁺/tear ♂×$Agcd^{A11}$ ♀ | CFP⁺/$cd^-$ | 20 | 20/20 (100%) | | | | |
| | CFP⁻/$cd^-$ | 3 | | | | 3/3 (100%) | |
| F1 CFP⁺/tear ♀×$Agcd^{A11}$ ♂ | CFP⁺/$cd^-$ | 20 | 20/20 (100%) | | | | |
| | CFP⁻/$cd^+$ | 3 | | 3/3 (100%) | | | |
| **F1 female lineage** | | | | | | | |
| F1 CFP⁺/$cd^+$ ♂×$Agcd^{A11}$ ♀ | CFP⁺/$cd^-$ | 60 | 60/60 (100%) | | | | |
| | CFP⁻/$cd^+$ | 59 | | | 59/59 (100%) | | |
| | CFP⁻/$cd^-$ | 27 | | | | 27/27 (100%) | |
| F1 CFP⁺/$cd^+$ ♀×$Agcd^{A11}$ ♂ | CFP⁺/$cd^-$ | 60 | 60/60 (100%) | | | | |
| | CFP⁻/$cd^+$ | 74 | | 41/74 (55%) | 33/74 (45%) | | |
| | CFP⁻/$cd^-$ | 33 | | | | 33/33 (100%) | |
| F1 CFP⁺/tear ♂×$Agcd^{A11}$ ♀ | CFP⁺/$cd^-$ | 60 | 60/60 (100%) | | | | |
| | CFP⁻/$cd^+$ | 5 | | | 5/5(100%) | | |
| | CFP⁻/$cd^-$ | 22 | | | | 22/22 (100%) | |
| F1 CFP⁺/tear ♀×$Agcd^{A11}$ ♂ | CFP⁺/$cd^-$ | 60 | 60/60 (100%) | | | | |
| | CFP⁻/$cd^+$ | 56 | | 20/56 (36%) | 36/56 (64%) | | |
| | CFP⁻/$cd^-$ | 23 | | | | 23/23 (100%) | |
| F1 CFP⁺/$cd^-$ ♂×$Agcd^{A11}$ ♀ | CFP⁺/$cd^-$ | 20 | 20/20 (100%) | | | | |
| | CFP⁻/$cd^-$ | 30 | | | | 28/30 (93%) | 2/30 (7%) |
| F1 CFP⁺/$cd^-$ ♀×$Agcd^{A11}$ ♂ | CFP⁺/$cd^-$ | 20 | 20/20 (100%) | | | | |
| | CFP⁻/$cd^+$ | 20 | | | 19/20 (95%) | 1/20 (5%) | |
| | CFP⁻/$cd^-$ | 30 | | | | 24/30 (80%) | 6/30 (20%) |

CFP⁺, cyan fluorescent protein positive; $cd^+$, black eye, wild-type *cardinal* allele; $cd^-$, red eye, nonfunctional mutant *cardinal* allele.
[a]Number and percent with genotype: D, drive allele CFP⁺; W, wild-type allele; R, functional NHEJ allele; B, nonfunctional NHEJ allele; B/B progeny are mosaic with 2 nonfunctional NHEJ alleles.
[b]Number of samples collected and sequenced from all replicates of each cross.

parent: 1.9% [39/2,064]; tear male parent: 1.6% [27/1,653]; tear female parent: 1.1% [24/2,093]), and except for the male parent tear cross, these too were higher (4- to 57-fold) than what was observed in the male founder lineage due to previous maternal effects that produced cleavage-resistant B alleles.

Testcrosses of the recovered female-lineage mosquitoes with the CFP⁺/$cd^-$ phenotype produced F2 progeny with GDIs that were not significantly different from a normal 50:50 meiotic segregation ratio in both male (50.6%, $X^2$= 0.024, df = 1, P-value > 0.1) and female crosses (52.7%, $X^2$= 0.74, df = 1, P-value > 0.1) (Fig. 2; Table 3). These data are consistent with the conclusion that the $cd^-$ alleles in the F1 parents are cleavage-resistant. The cGDI calculation is not applicable to these testcrosses because there were no wild-type target alleles in the germlines of CFP⁺/$cd^-$ hemizygotes available for HDR-mediated gene conversion. Furthermore, the small percentage, 5.5% (29/524), of black-eye mosquitoes found in the F2 progeny of the female testcross were shown subsequently to be cleavage-resistant R alleles present in mosaic germlines (Table 4).

## Genotypic analyses of F2 progeny of hemizygous AgTP13 testcrosses

The genotypes of 916 F2 progeny from the male- and female-lineage testcrosses were sequenced, 458 of them with expected phenotypes from the reciprocal CFP⁺/$cd^+$ × $Agcd^{A11}$ crosses and 458 with exceptional phenotypes that were the progeny of F1 parents with exceptional phenotypes (tear and $cd^-$) (Table 4). Individual samples sequenced are labeled "TC-xxxx" to indicate that they come from the testcrosses (Supplementary Tables 5 and 6).

### General sequence properties of mutant target-site alleles

As expected based on the design of the experiments, all 916 sequenced F2 progeny from both lineages carried the $Agcd^{A11}$ allele. Furthermore, all 440 CFP⁺ progeny from both lineages carried the expected D allele, while the 476 CFP⁻ samples carried single W (148), R (153), or B (167) alleles, or were heteroallelic for 2 B (B/B, 16) alleles, resulting in 336 total sequenced mutant alleles (Table 4). A total of 31 of the alleles were PAM⁺, 12 of which were in the PEPPR class (TC-610 and the clusters represented by TC-318/382/427/471/556/577/596/632/747/750; Supplementary Tables 5 and 6).

The B/B samples are enigmatic because their sequencing shows clearly that there are 2 distinct B alleles in addition to the expected $Agcd^{A11}$ allele, and they appear in clusters of 2 and 6 samples represented by TC-539 and TC-569, respectively (Supplementary Table 6). Another sample, TC-408, was scored phenotypically with a black eye ($cd^+$, W, or R allele), but the molecular genotype shows a −3/+17 bp indel mutation that disrupts the reading frame. Furthermore, this mutation is identical to the one in OC-1141 that was scored phenotypically as red eye ($cd^-$) and designated a B allele, and it is included in the B allele testcross total (Supplementary Tables 1 and 6). Accelerated nonenzymatic oxidation of eye pigment precursors may account for the eyes appearing dark in the TC-408 sample (see Materials and Methods; Li et al. 1999).

The majority, 97.6% (328/336) of R and B mutations sequenced were recovered in clusters ranging from 2 to 28 mosquito samples from independent replicate experiments (Supplementary Fig. 3; Supplementary Tables 5 and 6). For brevity, only 1 sequence from each cluster is referenced in the following text unless there

**Table 5.** Parameter values for calculating mutation rate/allele/generation in testcross F2 progeny.

| | Male | | | Female | | | Total | | |
|---|---|---|---|---|---|---|---|---|---|
| **Total alleles screened[a]** | | 7,138 | | | 8,565 | | | 15,703 | |
| **F1 CFP+/cd+ ♂/♀ and F1 CFP+/Tear ♂/♀ reciprocal crosses to Agcd^A11** | | | | | | | | | |
| Total alleles adjusted[b] | | 7,138 | | | 7,711 | | | 14,849 | |
| Total exceptional phenotypes scored[c] | | 108 | | | 404 | | | 512 | |
| Total exceptional genotypes sequenced[d] | | 97 | | | 299 | | | 396 | |
| Allele type sequenced[e] | W | R | B | W | R | B | W | R | B |
| | 87 | 1 | 9 | 61 | 133 | 105 | 148 | 134 | 114 |
| Independent alleles[f] | | 1 | 4 | | 7 | 12 | | 8 | 16 |
| Total independent alleles[g] | | 5 | | | 19 | | | 24 | |
| Total independent alleles adjusted[h] | | 5.567 | | | 25.67 | | | 31.0 | |
| Mutations/target gene/generation[i] | | 0.00077 (0.08%) | | | 0.00332 (0.33%) | | | 0.0021 (0.21%) | |
| **F1 CFP+/cd+ ♂/♀ reciprocal crosses to Agcd^A11** | | | | | | | | | |
| Total alleles adjusted[j] | | 6,517 | | | 3,965 | | | 10,482 | |
| Total exceptional phenotypes scored[c] | | 101 | | | 289 | | | 390 | |
| Total exceptional genotypes sequenced[d] | | 91 | | | 193 | | | 284 | |
| Allele type sequenced[e] | W | R | B | W | R | B | W | R | B |
| | 84 | 1 | 3 | 41 | 92 | 60 | 125 | 93 | 61 |
| Independent alleles[f] | | 1 | 3 | | 2 | 3 | | 3 | 6 |
| Total independent alleles[g] | | 4 | | | 5 | | | 9 | |
| Total independent alleles adjusted[h] | | 4.43 | | | 7.48 | | | 12.36 | |
| Mutations/target gene/generation[i] | | 0.00067 (0.07%) | | | 0.00188 (0.19%) | | | 0.00117 (0.12%) | |

[a]Total alleles screened (Table 3).
[b]Total alleles adjusted does not include data from the reciprocal F1 CFP+/cd⁻ × Agcd^A11 testcrosses (Table 3).
[c]Total exceptional phenotypes scored (Table 3).
[d]Total exceptional genotypes sequenced (Table 4, Supplementary Tables 5 and 6).
[e]Allele type sequenced: W, wild-type; R, functional mutant allele; B, nonfunctional mutant allele (Table 4, Supplementary Tables 5 and 6).
[f]Independent alleles include only those arising in the F2 generation and not inherited (Supplementary Tables 5 and 6).
[g]Total independent alleles (Supplementary Tables 5 and 6).
[h]Total independent alleles-adjusted = total exceptional phenotypes scored/total exceptional genotypes sequenced × total independent alleles (Tables 3 and 4).
[i]Mutations/target gene/generation = total independent alleles-adjusted/total adjusted alleles.
[j]Does not include data from the reciprocal F1 CFP+/cd⁻ or CFP+/Tear ♂/♀ × Agcd^A11 testcrosses (Table 3).

is a reason to list them in full. A single cluster in a replicate cross most likely represents multiple progeny inheriting the identical allele from the products of a single spermatogonial or oogonial cell in the germline of a male or female parent, respectively, and these were grouped accordingly in Supplementary Tables 5 and 6. Taking this into consideration, a total of 51 distinct mutant genotypes, 16 R and 35 B, were identified, with 13 and 29 of the R and B alleles, respectively, recovered in clusters. As was seen with the sequences of F1 outcross progeny, it is possible to identify 23 of the mutations as most likely arising from NHEJ, 20 from MMEJ and 8 with the previously described characteristics of alt-MMEJ.

### Specific properties of R target allele mutations

Similar to the F1 outcross progeny, all R alleles either restore the ORF with deletions of 6, 9, 12, or 15 bp, or with combinations of compensatory mutations (e.g. TC-421 [−1, +1 bp], the TC-422 cluster [−3, +3 bp], the TC-402 cluster [−15, +6 bp], and TC-406 [−8, +2 bp]) (Supplementary Table 5). Three R allele sequences were recovered in multiple distinct replicates, TC-281/301/463 (−9 bp), TC-318/382/427/471 (−6 bp), and TC-426/661 (−6 bp). Eight of the R allele sequences arose from NHEJ and 8 from MMEJ.

Distinguishing mutant R alleles created de novo in the F1 parental germlines from those inherited vertically through the parents can be done using the data in Supplementary Tables 2 and 5 and focusing on the specific lineage relationships, independent replicates, and sex of the gene-drive donor parents. Five of the 16 R allele genotypes, TC-401/402/406/421/422, are seen only in the F2 generation progeny (Supplementary Tables 2 and 5). However, the 2 genotypic clusters represented by TC-401 and TC-402 and the single sample from TC-406 come from the CFP+/cd⁻ testcrosses and were likely inherited from an ovary mosaic for R and B alleles to account for the black-eye progeny recovered

from these crosses (Table 3). Furthermore, the fact that TC-401 and TC-402 represent clusters that contributed to the near 50:50 segregation patterns of the crosses from which they were derived provides additional support for this interpretation. Therefore, only 2 of the R allele genotypes, TC-421/422, seen only in the F2 generation progeny, result from germline mutation events. However, while the 6 genotypes, TC-661/281/318/427/463/471, were seen in F1 parents, they appear in F2 progeny derived from unrelated crosses (Supplementary Tables 2 and 5). So, while the genotypes are identical, the crossing schemes support the conclusion that these were created de novo in the respective F1 parental germlines, yielding a total of 8 R germline mutations generated de novo.

Five of the R genotypes, TC-301/349/362/382/426, were observed in both F1 outcross parents and clusters of their corresponding replicate F2 progeny, and this supports the conclusion that these were inherited through the parental lineages. Combining these with the 3 from the CFP+/cd⁻ testcrosses yields 8 R genotypes inherited through the germlines of the F1 outcross parents.

### Specific properties of B target allele mutations

All 35 B alleles encode nonfunctional proteins resulting from disruptions of the ORF or the introduction of a stop codon (TC-577 cluster) (Supplementary Table 6). Four B allele sequences were recovered in multiple distinct replicates, TC-479 cluster/612 cluster (−14 bp), TC-511 cluster/555 cluster (−2, +13 bp), TC-629/571 cluster/634 cluster/746 (−11 bp), and TC-506 cluster/610/747 cluster (−1 bp). Fifteen of the sequences arose from NHEJ, 5 of which were indels, 12 had the characteristics of MMEJ, and the remaining 8 were alt-MMEJ events. All B sequences present in multiple samples were putative MMEJ events. The 8 alt-MMEJ alleles were

represented once in TC-555 and TC-408 and the clusters TC-511/542/569/577/601/615. Four of the multiple MMEJ clusters with identical 11-bp deletion sequences (TC-571/629/634/746) and 2 with 14-bp deletions (TC-479/612) were seen in the genotypes of the outcross F1 progeny and in other experiments performed in our laboratory (Supplementary Tables 2 and 3).

Using analyses similar to those of the R alleles, specific lineage relationships, independent replicates, and sex of the gene-drive donor parents, the data in Supplementary Tables 2 and 6 can be used to determine the mutant B alleles created de novo in the F1 parental germlines and those that are inherited. Twenty of the 35 B genotypes are seen only in the F2 generation progeny; however, the 11 from CFP$^+$/$cd^-$ testcrosses are likely inherited from either nonmosaic or mosaic parental gonads generated by maternal effects in the outcrosses, and this leaves 9 genotypes unique to the F2 progeny (TC-599/482/511/601/615/621/632/750/755; Supplementary Table 6). Furthermore, the structuring of the crosses supports the conclusion that 1 of the 14-bp deletions (TC-612), 3 of the 11-bp deletions (TC-746/629/634), and all 3 of the 1-bp deletions (TC-506/610/747) were created independently in the germlines of the F1 parents and this yields a total of 16 B genotypes that arose as independent germline events. Two of the remaining 19 genotypes (TC-479 and TC-630) had clear F1-F2 lineage relationships, and the remaining 17 were generated likely by maternal effects in the outcrosses and inherited through the CFP$^+$/$cd^-$ testcrosses.

## Germline mutation rates

The testcross sequencing data permit derivation of estimates of the rate per generation at which the Cas9/gRNA activity results in germline-generated mutations of wild-type target alleles in the hemizygous F1 parents (Tables 3 and 4, Supplementary Tables 5 and 6). However, female-lineage CFP$^+$/$cd^-$ male and female testcrosses contributed 854 (number of CFP$^+$/$cd^-$ progeny) of the 15,703 (number of total progeny) potential target alleles, and while we know that the samples they came from were phenotypically $cd^-$, the genotypes were a mixture of a large number of B alleles and a much smaller number of R alleles (Table 3, Supplementary Tables 5 and 6). Furthermore, the near-Mendelian segregation ratios in the outcomes of the testcrosses support the conclusion that all of the R and B alleles generated were cleavage-resistant; therefore, no additional mutant alleles are expected to result from them, and they are not considered further in the germline mutation rate estimations.

A total of 512 potential target alleles in the male- and female-lineage F1 CFP$^+$/$cd^+$ and CFP$^+$/tear testcrosses were not converted via HDR, and of the 396 sequenced, 148 were wild-type (W), 134 were R, and 114 were B (Tables 3 and 4, Supplementary Tables 5 and 6). As described above, exclusion of those alleles from the outcrosses that were inherited through the F1 parents and only counting 1 representative from each of the observed clusters, results in a total of 24 (8 R and 16 B) mutant alleles generated following independent NHEJ, MMEJ, and alt-MMEJ germline events during the production of F2 generation progeny. Not surprisingly, the number of R alleles is less than that of the B alleles, and the one-third observed is consistent with the fraction needed to conserve the ORF to get the black-eye phenotype (Tables 3, Supplementary Tables 5 and 6). Furthermore, while no significant differences were found between the numbers of independent mutations generated in the male and female F1 parental testcrosses from the male- and female-lineage outcrosses (3 vs 2 and 10 vs 9, respectively, Supplementary Table 7), there was a significant difference between the pooled numbers of the male and female

lineages, 5 vs 19, propagated through the 2 generations, F1 and F2, examined here ($X^2 = 8.167$, df = 1, P-value = 0.0043).

The 396 sequenced alleles represent ~77% of the total 512 nonconverted target alleles, 2.7% of the adjusted total 14,813 target alleles, and their genotypic complexity and ratios support the conclusion that they are representative of the total (Table 5). Extrapolating from the observed data that showed 24 independent mutant alleles occurring in the F2 generation, we expect the total number of independent mutant alleles generated from the total potential target alleles to be 31. Using this number and the adjusted population size yields a rate of ~0.21% mutations/target gene/generation. The same logic can be applied to estimate the average separate outcomes of the male and female lineages, ~0.08% and 0.33% mutations/target gene/generation, respectively. Furthermore, comparing the adjusted total number of independent alleles shows that ~80% result from mutagenic activity in female lineages.

Further restricting the analyses to only the outcomes of F1 CFP$^+$/$cd^+$ reciprocal testcrosses (excluding the data from both the CFP$^+$/$cd^-$ and CFP$^+$/tear testcrosses), yields values of ~0.07% and ~0.19% mutations/target gene/generation from the respective male and female germline lineages with a combined total of ~0.12% mutations/target gene/generation (Table 5). In the absence of the CFP$^+$/tear testcrosses, the contribution of the total adjusted independent alleles by the female lineage drops to ~60% of the total alleles generated.

## Paternal- and maternal-effect contributions to inherited mutations

The complexity of the genotypes in the progeny of the outcrosses and the sequencing approach used here prevent accurate estimates of the rates at which mutant alleles are generated by paternal and maternal effects in the F1 embryos. However, the most significant impact on drive system dynamics likely results from those that are actually inherited through the germline to the next generation (Supplementary Tables 2, 5, and 6). Having estimated above that 24 of the 51 distinct sequenced R and B alleles likely arose de novo in the germlines of the F1 progeny, the remaining 27 originated as paternal- or maternal-effect mutations that were inherited by the F2 progeny through the germline of the F1 parents. Thus, while we cannot estimate the rate of their formation, we can say that they account for slightly more than one-half, ~53%, of all mutant alleles recovered in the testcrosses.

## Cleavage-resistant alleles

The combined data can be used to estimate the frequency of inheritance of potential cleavage-resistant target-site mutations. A total of 31 of the 51 combined multiple, inherited, and independently originating alleles in the representative sequenced F2 progeny were PAM$^+$. The remaining 20 had partial or complete ablations of the PAM site (PAM$^-$), and it is reasonable to assume that these alleles would be resistant to Cas9/gRNA cleavage and therefore have a potential negative impact on the gene-drive system conversion efficiency. Furthermore, only 5, 1 R (TC-421 [−1, +1 bp indel]) and 4 B (the TC-747, TC-506, and TC-632 clusters and single TC610 sample [−1-bp deletion]) of the 31 PAM$^+$ mutant alleles had lesions small enough to allow possible HDR-mediated gene conversion. Previous work with the AgNosCd-1-based gene-drive system showed efficient cleavage of 4 of 5 target sites with SNPs, with the 1 exception being a target allele with a mutation in the first of the 3 nucleotides of the PAM site, which was cleaved, but not as efficiently (Carballar-Lejarazú et al. 2020). Therefore, it is reasonable to conclude that a minimum combined

90% (46/51) of the sequenced mutant alleles in the F2 progeny generation are gene-drive resistance alleles that can be expected to have an effect on gene-drive dynamics, and the impact may be greater if the single-nucleotide deletions have some level of cleavage-resistance.

## Cluster effects on inheritance

Cluster inheritance of mutant alleles has the potential to have a major impact on the overall drive system dynamics. Details of mosquito germline biology and early embryonic development provided in Supplementary Text 1 explain how a single mutagenic event can be amplified so that the ensuing mutation is over-represented in the progeny of the next generation. The testcross results identified 51 distinct paternal- and maternal-effect and germline R and B mutations distributed unequally among 336 sequenced progeny (Supplementary Tables 4 and 5; Supplementary Fig. 3). The cluster effects were greater in female lineage crosses where they amplified by ~7-fold the number of inherited R and B alleles (326) over the number of distinct mutations (46) compared to a 2-fold effect seen in males (10/5) (Supplementary Table 8). It is important to emphasize that while the fold values derived here are robust in the context of this specific work, they should not be used directly to estimate inheritance frequencies in gene-drive experimental trials. This is because the testcrosses were designed to estimate mutation rates among different phenotype-based matings and aggregating them as done here is not a likely representation of the complexity and frequency of each mating combination that would occur in a mixed population of phenotypes and genotypes.

## Discussion

Multiple factors complicate determining rates of gene-drive system-mediated mutagenesis in mosquitoes (Carballar-Lejarazú et al. 2020). These include the long-term stability of Cas9/gRNA complexes, paternal and maternal effects in embryos on target alleles derived from nondrive-carrying parents, mutagenic events at multiple developmental stages in hemizygous drive-carrying diploid cells that give rise to clonal clusters in testes and ovaries that amplify low mutation rates into high heritability rates, and cryptic genotypic mosaicism resulting from the use of tissue-restricted marker genes and nuclease expression. Some of these factors, for example, the clustering effects, result from the biology of gametogenesis in adult males and females (Supplementary Text 1), and others result from the specific design features of the drive systems themselves and their analyses (e.g. recessive marker genes visible only in the eye, sequencing strategy). We used a simple, 2-step mating scheme to characterize this complexity for AgTP13, a strain designed for population modification of the African malaria mosquito, *An. gambiae*.

The first step was to outcross homozygous, gene-drive males and females in separate lineages to a reference strain and characterize the resulting phenotypes and genotypes. The integrated gene-drive construct itself represents a cleavage- and drive-resistant allele; therefore, all wild-type target alleles come from the nondrive carrying parents. Furthermore, since the drive-carrying parents are homozygous, no Cas9/gRNA-mediated events occur in their germlines, and all activity results from either paternal or maternal effects in the ensuing embryos following fertilization. Yeast nucleases and Cas9/gRNA complexes have been interpreted to be remarkably stable in insects and capable of vertical transfer through both sperm and eggs to explain results from crossing data (Windbichler et al. 2008; Gantz et al. 2015; Guichard et al. 2019; Carballar-Lejarazú et al. 2020, 2022; Auradkar et al. 2024; Meccariello et al. 2024). This

inherited Cas9/gRNA-mediated activity occurs in mosquitoes at a much higher frequency in the progeny of outcrosses of drive-carrying females than in the reciprocal male crosses (Gantz et al. 2015; Carballar-Lejarazú et al. 2020, 2023).

Sanger sequencing of the gRNA target sites in F1 progeny from mosquitoes with expected and exceptional phenotypes revealed a complexity of mutations, some of which could be identified easily, while others were classified as cryptic or disturbed. Careful examination of the "clean" target allele sequences from mosquitoes with exceptional phenotypes showed approximately equal representation of NHEJ and MMEJ alleles (~36% each) and a smaller proportion, 26%, of alt-MMEJ allele. The majority (84%) of MMEJ alleles appeared more than once as independent events in different samples. No evidence of postcleavage SSA was seen. The unresolved genotypes result from extensive somatic mosaicism, and it might be possible to determine them using next-generation sequencing methods. While this somatic mosaicism may have an impact on the fitness of individual mosquitoes ("mosaic lethality" as described by Guichard et al. 2019), resolving the individual mutations was not needed for this study.

The second step was to set up testcrosses that would allow direct estimates of the formation of mutations as a consequence of Cas9/gRNA activity in the germlines of hemizygous, drive-carrying parents. The male and female parents derived from the F1 progeny of the 2 lineage outcrosses were mated to the homozygous $Agcd^{A11}$ mutant strain. Since $Agcd^{A11}$ is a cleavage- and drive-resistant allele, all potential target alleles are present in the germlines of the hemizygous, drive-carrying parents. Molecular genotyping of the F2 progeny revealed similar average distributions of ~42% each for NHEJ and MMEJ alleles, with the alt-MMEJ alleles representing a smaller fraction, ~16%, than what was seen in the outcross F1 progeny, again with no evidence of SSA. The design of the experiments allowed an estimate of the Cas9/gRNA-mediated germline mutation rate/target gene/generation, which was estimated to be 0.21% with ~80% of these arising in the female lineage. Furthermore, these data allow estimating the percentages of inherited mutations that arose independently in the parental germlines (47%) and those that are inherited through paternal and maternal effects (53%).

It is difficult to explain the recovery of testcross samples with multiple B alleles (B/B). Polyandry (1 female mating with 1 or more genetically distinct males) has been observed in both field and laboratory anopheline populations (Mahmood and Reisen 1980; Tripet et al. 2003). Invoked here as a possible cause for the multiple alleles, then requires polyspermy, multiple sperm entering a single oocyte. Polyspermy has also been observed microscopically in mosquitoes (reviewed by Degner and Harrington 2016), although it is expected to lead to unstable polyploidy and aneuploidy and ultimately cell death. However, this explanation is confounded further by the recovery of all B/B genotypes in clustered samples. We are reluctant to attribute the results to experimental error, considering the diligence with which the work was carried out, and follow-up investigations by us and others may reveal some aspect of the biology that we are missing.

A number of efforts have been made to derive germline mutation rates in diverse mosquito species for both population suppression and replacement/modification strategies (Gantz et al. 2015; Champer et al. 2017; Hammond et al. 2017, 2021a,b; Fuchs et al. 2021; Carballar-Lejarazú et al. 2022; Anderson et al. 2024; Yang et al. 2025). However, direct comparisons of our results with these studies are complicated by the details of the experimental designs. One study in a different mosquito species, the yellow fever mosquito *Aedes aegypti*, had a similar design, outcrosses

of homozygous drive-carrying males and females to wild types, followed by testcrosses of hemizygous drive-carrying progeny to a homozygous drive-resistant mutant strain (Xu et al. 2025). However, the specifics of the design of the gene-drive system (target gene choice and promoters used to express the Cas9 nuclease and gRNA) and subsequent performance (low cGDIs) prevent a meaningful comparison of the results of this work with those of our study. The average rates of formation of lineage-specific germline mutations in our work, 0.08% in males and 0.33% in females, are 45- to 61-fold lower than the respective values reported for *Ae. aegypti*. A major contributor to these differences is likely to be the exceptionally high rates of HDR/cGDI seen in AgTP13 that result in the conversion of target alleles to cleavage-resistant D alleles that prevent the formation of any other mutant alleles.

The multigenerational female Cas9/gRNA-mediated mutagenesis bias propagated through the F1 to the F2 generation seen here was described first in work on the vinegar fly, *Drosophila melanogaster* (Guichard et al. 2019). This effect, named "shadow drive," was seen subsequently in mosquitoes and in additional work with *D. melanogaster*, where it was seen to contribute in most cases to lower drive efficiencies, most likely as the result of the production of cleavage-resistant target alleles (Adolfi et al. 2020; López Del Amo et al. 2020; Li et al. 2021, 2024; Auradkar et al. 2024). The results presented here further support the long-term stability of Cas9/gRNA systems in insects.

We present these data to allow comparisons with what will be found in other insect gene-drive systems, and in particular, to better inform modeling of the consequences of drive-generated mutations on subsequent spread dynamics of these systems in mosquito populations (Hammond et al. 2016; Unckless et al. 2017; Sánchez C et al. 2020). Previous modeling has shown favorable outcomes for the AgNosCd-1-based drive systems due mainly to the high drive efficiency and low impact on fitness resulting from drive-system insertions (Lanzaro et al. 2021; Carballar-Lejarazú et al 2023). It is important to note that the presented calculated values here are likely specific to the AgTP13 laboratory strain. Similar analyses are recommended for potential release strains with different gene-drive systems and those in recently colonized wild-type genetic backgrounds. Hopefully, the work here serves as a framework for future assessments of such release strains.

## Data availability

The datasets generated and/or analyzed during the current study are available in the figures, tables, and Supplementary Material. Plasmids and mosquito lines are available from the authors on request.

Supplemental material available at GENETICS online.

## Acknowledgments

We are grateful to Drusilla Stillinger, Devin Nguyen, and Lorena Winokur for mosquito husbandry and to Bryn Hobson for assistance in developing some of the figures.

## Funding

The University of California Irvine Malaria Initiative (UCIMI), the Bill and Melinda Gates Foundation (INV-043645), National Institutes of Health (AI170692), and an anonymous donor. AAJ is a Donald Bren Professor at the University of California, Irvine.

## Conflicts of interest

None declared.

## Author contributions

RC-L, TBP, TT, and AAJ designed experiments; RC-L, TBP, and TT conducted experiments; RC-L, TBP, TT, and AAJ analyzed data; and RC-L, TBP, TT, and AAJ prepared the manuscript.

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
