## [Peer Review File · Genetics]

Mutant allele formation and inheritance during Cas9/guide RNA-mediated gene drive in a population modification mosquito strain for human malaria control

Rebeca Carballar-Lejarazú, Thai Pham, Taylor Tushar, and Anthony James

NOTE: The reviews and decision letters are unedited and appear as submitted by the reviewers. In extremely rare instances and as determined by a Senior Editor or the EIC, portions of a review may be redacted. If a review is signed, the reviewer has agreed to no longer remain anonymous. The review history appears in chronological order.

Review Timeline:

Submission Date:	2025-06-17
Editorial Decision:	2025-07-14
Resubmission Received:	2025-08-11
Accepted:	2025-08-15

In the manuscript titled “ *Mutant allele formation and inheritance during Cas9/guide RNA-mediated gene drive in a population modification mosquito strain for human malaria control*”, the authors present a detailed experimental investigation into the formation and inheritance of resistance alleles in a CRISPR-Cas9-based gene drive system (TP13) designed for population modification of *Anopheles gambiae*, a key vector of human malaria. Through a series of extensive outcross and testcross experiments, the authors estimate the frequency, origin, and impact of Cas9-induced mutations at the target site in the mosquito genome, with implications for gene drive stability, spread, and possible field deployment. The authors have clearly and extensively characterized the maternal versus paternal influences on resistance allele development. The data demonstrate that maternal effects dominate, accounting for ~80% of de novo germline mutations and a disproportionate percentage of inherited Cas9/gRNA-induced mutations caused by deposition into oocytes. Furthermore, the authors give demonstrate clustering effects in both sexes, where mutations occurring in early germline stem cells or syncytial embryos are overrepresented in progeny due to clonal amplification. The meticulous genotyping of over 900 people and the classification of mutant allele types (NHEJ, MMEJ, alt-MMEJ) adds depth and mechanistic clarity.

This study will be valuable to the field particularly as it provides critical empirical parameters for future gene drive modeling and risk assessment.

I, however, has few minor concerns:

1. The manuscript is too descriptive, especially the result section. It will be better to rearrange some of the descriptive data to graphs. As is, the manuscript is very difficult to follow and engage with.
2. The existing figures could use clearer nomenclature to allow readers to follow what section of the figure is being discussed in the text.

July 14, 2025

GENETICS-2025-308285

Mutant allele formation and inheritance during Cas9/guide RNA-mediated gene drive in a population modification mosquito strain for human malaria control

Dear Dr. James:

Several experts in the field have reviewed your manuscript, and I have read it as well. I am pleased to inform you that, with minor revisions, it is potentially suitable for publication in GENETICS. The reviewers have comments and concerns that need to be addressed in a revised manuscript. You can read their reviews at the end of this email.

It is most important that you address the following in your resubmission: Please significantly revise the manuscript for clarity following all three reviewers' suggestions, which includes moving results to tables or figures rather than including an extensive description of the results in the text (Reviewers 1 and 3); address the questions and suggestions about sequencing approaches raised by Reviewers 2 and 3; address the concern about whether the data support the "threshold hypothesis" (Reviewer 2), and discuss the findings presented in this manuscript in relation to the work of other groups (Reviewer 2).

We look forward to receiving your revised manuscript. Please let the editorial office know approximately how long you expect to need for revisions.

Upon resubmission, please include:

1. A clean version of your manuscript;
2. A marked version of your manuscript in which you highlight significant revisions carried out in response to the major points raised by the editor/reviewers (track changes is acceptable if preferred);
3. A detailed response to the editor's/reviewers' comments and to the concerns listed above. Please reference line numbers in this response to aid the editors.

Additionally, please ensure that your resubmission is formatted for GENETICS.

<https://academic.oup.com/genetics/pages/general-instructions>

Follow this link to submit the revised manuscript: Link Not Available

Sincerely,

Jill Wildonger
Associate Editor
GENETICS

Approved by:
Kate O'Connor-Giles
Senior Editor
GENETICS

Reviewer #1 :

In the manuscript titled " Mutant allele formation and inheritance during Cas9/guide RNA- mediated gene drive in a population modification mosquito strain for human malaria control", the authors present a detailed experimental investigation into the formation and inheritance of resistance alleles in a CRISPR-Cas9-based gene drive system (TP13) designed for population modification of *Anopheles gambiae*, a key vector of human malaria. Through a series of extensive outcross and testcross experiments, the authors estimate the frequency, origin, and impact of Cas9-induced mutations at the target site in the mosquito genome, with implications for gene drive stability, spread, and possible field deployment. The authors have clearly and extensively characterized the maternal versus paternal influences on resistance allele development. The data demonstrate that maternal effects dominate, accounting for ~80% of de novo germline mutations and a disproportionate percentage of inherited Cas9/gRNA-induced mutations caused by deposition into oocytes. Furthermore, the authors give demonstrate clustering effects in both sexes, where mutations occurring in early germline stem cells or syncytial embryos are overrepresented in progeny due to clonal amplification. The meticulous genotyping of over 900 people and the classification of mutant allele types (NHEJ, MMEJ, alt-MMEJ) adds depth and mechanistic clarity.

This study will be valuable to the field particularly as it provides critical empirical parameters for future gene drive modeling and risk assessment.

I, however, has few minor concerns:

1. The manuscript is too descriptive, especially the result section. It will be better to rearrange some of the descriptive data to graphs. As is, the manuscript is very difficult to follow and engage with.
2. The existing figures could use clearer nomenclature to allow readers to follow what section of the figure is being discussed in the text.

Reviewer #2 :

The authors present a thorough investigation into how parental deposition of Cas9 gene drive components into the fertilized embryo can result in somatic mosaicism in *An. gambiae* mosquitoes, and furthermore to what extent does this mosaicism extend to the germline and become transmitted to subsequent generations. This publication is timely, because as gene drive technologies continue to develop it is imperative that generation of target site mutations that will cause resistance to the drive are accounted for or avoided.

The publication's strategy was clearly communicated, figures were easy to follow, and the conclusions were backed up by data presented. However, as it is currently written it will be penetrable by only a few gene drive experts and even then, it will take some effort to parse out the results. In several places the text could benefit from a short sentence or two to help non-experts understand rationale and outcomes. There are about 270 lines of results that exhaustively describe all findings but often a structure is lacking. There are virtually no sub-headings and where there are they are not very informative to the reader with regards to findings or approach e.g "Inheritance patterns and phenotypic and genotypic analyses of hemizygous AgTP13 testcrosses " While this might sound onerous or daunting I don't think it's a hard ask - the Discussion is well written and has a structure, including expansion of the logic behind the experimental design, that would lend itself to the restructuring of the results. Another example, when performing the testcrosses (Figure 2) it is not immediately obvious that one objective is to assess how frequently mutations giving rise to mosaicism in a parent are transmitted to offspring. This example goes on to be an important point in the discussion, but is unclear in the relevant results section.

Some of the previous work, in this area of understanding maternal and paternal contributions to Cas9-induced mutations that affect homing, to be undercited. In particular the work of the groups of Chamber, Crisanti, Gantz/Bier. Indeed in the conclusion it is stated that "We present these data to allow comparisons with what has been found in other insect gene-drive systems" but this wasn't really done here.

We note that the authors use the nomenclature 'R' and 'B' for functional and non-functional target site mutations, whereas other literature refers to these as 'R1' and 'R2', respectively. We acknowledge that R and B may be clearer to the reader of this publication, but suggest that it may be beneficial to keep nomenclature consistent with other works.

Similarly. The use of the term cGDI is confusing and not helpful - this would be referred to as homing rate by most other researchers in the field. It then also does not make sense to refer to them as if they were independent e.g. line 267 "The difference between the GDI and cGDI could be explained by.." - for any value in which GDI is not 100% the GDI and cGDI will always be different - they are inextricable. So on line 272 "GDI, 96.8%, and cGDI, 93.6% " are derived because $(96.8-50)/50$ is 93.6. The authors know this of course but to the uninitiated it could read as if they were independent outcomes.

The authors describe that when molecular genotyping was carried out using Sanger sequencing that 82 F1 samples were unresolved, and labelled 'cryptic' or 'disturbed'. Given that 587 F1 genotypes were successfully accessed via Sanger sequencing, it seems a shame that the authors stopped short of investigating the remaining 82 via another approach that may yield greater resolution - such as amplicon sequencing. Indeed, the Sanger sequencing described here already makes use of PCR-based enrichment, so amplicon sequencing of these samples should be relatively straightforward. While not essential for publication, investigation of these unresolved genotypes would benefit the work and potentially reveal additional mutations.

Line 452 - change the term horizontal transfer as this could be confounded to mean horizontal transfer - i.e. between non-siblings or even between species, which is not the intended meaning of the authors

Other Minors:

A couple of typos were spotted:

- Line 154 - 'Dreamtag' PCR Mix, should presumably be 'Dreamtaq'
- Line 533 - 'Hammond et al. 2026', presumably 2016
-

Line 49 - SSA is a form of homology-directed repair therefore the positioning of it as 'rather' is awry. Authors should specify HDR using the homologous chromosome as template

Lines 512-524 the argument that "these results support a hypothesis that a threshold level of introduction must be achievedto

mitigate resistance". Two things of note here: 1-the rate of accumulation per non-homing event (i.e. not per offspring) is likely the same between the two species and these observable rates allow a calculation of the propensity for resistance to arise 2- it doesn't make sense to me to contain one's thinking to the goal of suppressing a cage - at the edge of a population (or release area) there will always be areas where the drive is at low frequency, and therefore the proposed mitigation would not apply here, since resistance alleles would arise and backfill. Therefore, as written, I do not see how the data support the threshold hypothesis - at least not as a linear path to predicting an operational optimum for a strain that accumulates blocking resistance alleles in a cage setting.

Reviewer #3 :

Carballar-Lejarazu et al present and analysis of new Cas9-resistant alleles generated from an engineering homing-based gene drive transgene. Two important findings are noted. First, the large-scale test cross and sequencing performed here allows the authors to establish a clear rate of resistance allele formation to an accuracy that could not have been obtained without this specific test cross. Second, the authors identify a complex mixture of deletions and show that both in-frame and out-of-frame deletions are generated. While much of the manuscript is well-written, certain sections of the results section are dense and the author's decision to not express any of their findings visually (it is all tables/supplemental tables) makes it more challenging to extract key findings and concepts than it should be.

The researchers performed extensive Sanger-based sequencing on a large number (587) of F1 individuals (presented in Table 2). The rationale for this is not made clear, as this is essentially a characterization of somatic mutations, while the purpose of the study is to characterize germline mutations that impact Cas9-based homing (data later in the study gets at this more important question). The choice of Sanger sequencing is also quite strange, since this technology can only really give interpretable data when there is a single predominant sequence (or at most two equally proportional alleles). Somatic disruptions are almost always likely to be complex and incomplete. Thus, it is not surprising that almost all of the mosaic individuals sequenced produced no meaningful data (either no difference from WT due to frequency of somatic mutations being too low, or too many different disruptions). Really, the authors should have used a high-throughput method (ONT or Illumina) if their objective was to actually characterize these somatic events. But again, there doesn't seem to be a clear rationale for characterizing them in the first place.

The end result is only 53 deletion events, presented in detail in Tables/supplemental tables (using ONT or illumina on even 1 individual would probably reveal as many or more events, again reinforcing that the authors effectively produced very little data despite great effort). I was surprised there was no figure that summarized these events, since that would make it much easier to observe patterns/relationships. Instead, the authors walk us through a very long description of the numbers of each type, which is difficult to follow.

Finally, we get to the real experiment, test-crossing F1 individuals to the AgcdDEL11 test strain, which will uncover new germline-based target site mutations. Here, Sanger-based sequencing is perfectly appropriate, since each individual will only inherit a single new disrupted allele. However, as before, the authors do not synthesize the results into any figures to help the reader understand the main findings. Instead, there is a long and difficult to follow description of various classes/categories with continuous reference to supplemental tables.

Lines 324-326, the authors assume that "A single cluster in a replicate cross most likely represents multiple progeny inheriting the identical allele from the products of a single spermatogonial or oogonial cell in the germline of a male or female parent, respectively". However, the authors performed pooled crosses, so this approach will underestimate the rate of occurrence of common deletions and overestimate the rate of rare events. I could not find in the methods section any mention of the size of each cross, so it is impossible to determine the expected scale of this distortion (big difference between 5M x 5F and 50M x 50F, the latter would be expected to have many more common events). The assumption made by the authors would be valid only if they had performed single pair crosses (which would have been a much more powerful dataset), but they did not. This needs to be clearly stated.

16 samples contained two different mutant alleles in addition to the expected DEL11 test cross allele. This is fascinating and could be the result of fertilization with more than 1 sperm. Is it known how often this occurs in Anopheles?

The overall mutation rate estimates are important findings, and would be better represented in the main manuscript rather than the supplement (Table S8).

Not sure why the term "tear" is used in place of mosaic. The latter term is widely understood by the genetics community and the change thus makes the manuscript harder to follow.

The discussion can be streamlined as there is a duplicative re-presentation of the results (Lines 457-474)

Associate Editor Comments:

Please add Figure S4 to the list of Supplemental Materials.

Responses to comments of the reviewers

Reviewer #1 :

...minor concerns:

1. The manuscript is too descriptive, especially the result section. It will be better to rearrange some of the descriptive data to graphs. As is, the manuscript is very difficult to follow and engage with.

All three reviewers commented on the difficulty of following the narrative because of the detail used to describe the findings in the Results section. What we have done in response to these comments is simplify the presentation of some of the data, and importantly and in response to the suggestion of Reviewer #2, headings and subheadings were added to make it easier to follow the data presentation. We feel that the detailed description is appropriate for the first-time presentation of such data for this gene-drive approach and in keeping with this being a contribution to the genetics primary literature. We hope the editing and various headings and subheadings make it easier to follow the organization of the presentation of the results without sacrificing the details necessary for a relatively complex genetic analysis.

2. The existing figures could use clearer nomenclature to allow readers to follow what section of the figure is being discussed in the text.

Thank you for pointing this out. While we feel that the nomenclature itself is relatively clear, what was missing was an adequate description in the legends of what was being depicted and appropriate referencing in the text. The figure legends were edited to add more detail explaining what was being shown in the figures and referencing in the text was increased.

Reviewer #2 :

... as ... currently written it will be penetrable by only a few gene drive experts and even then, it will take some effort to parse out the results. In several places the text could benefit from a short sentence or two to help non-experts understand rationale and outcomes.

This is a good suggestion, and the text was revised to include at each heading and subheading what the ensuing paragraphs were communicating.

There are about 270 lines of results that exhaustively describe all findings but often a structure is lacking. There are virtually no sub-headings and where there are they are not very informative to the reader with regards to findings or approach e.g "Inheritance patterns and phenotypic and genotypic analyses of hemizygous AgTP13 testcrosses " While this might sound onerous or daunting I don't think it's a hard ask –

This is an excellent suggestion and headings and subheadings were added to emphasize the organization, structure and subject matter of the ensuing text.

Another example, when performing the testcrosses (Figure 2) it is not immediately obvious that one objective is to assess how frequently mutations giving rise to mosaicism in a parent are transmitted to offspring. This example goes on to be an important point in the discussion, but is unclear in the relevant results section.

Acknowledged and text was added to explain this.

Some of the previous work, in this area of understanding maternal and paternal contributions to Cas9-induced mutations that affect homing, to be undercited. In particular the work of the groups of Chamber, Crisanti,

Gantz/Bier. Indeed in the conclusion it is stated that "We present these data to allow comparisons with what has been found in other insect gene-drive systems" but this wasn't really done here.

*At the time of this writing, there is only one other study in mosquitoes with a similar design, outcrosses of homozygous drive-carrying males and females to wild-types, followed by testcrosses of hemizygous drive-carrying progeny to a homozygous drive-resistant mutant strain, and this was done with the yellow fever mosquito, *Aedes aegypti* (Xu et al., 2025) and we cite that. As we noted in the Discussion, 'the specifics of the design of the gene-drive system (target gene choice and promoters used to express the Cas9 nuclease and gRNA) and subsequent performance (low cGDI) prevent a meaningful comparison of the results of this work with those of our study.) However, the reviewer is correct in pointing out that others have made efforts to derive mutation rates using different species and experimental designs and we now include references to this work to acknowledge this work.*

We note that the authors use the nomenclature 'R' and 'B' for functional and non-functional target site mutations, whereas other literature refers to these as 'R1' and 'R2', respectively. We acknowledge that R and B may be clearer to the reader of this publication, but suggest that it may be beneficial to keep nomenclature consistent with other works.

We understand the point. Unfortunately, the genetics literature abounds in its use of "R" to mean many different things with 'resistance' of one form or another being the most predominant. In mosquito molecular genetics work, it was co-opted by some modelers to represent drive-resistant target alleles (unfortunately, absent in many cases of any empirical data to justify this designation). We adopted the parameter definitions of R and B used by the modelers with whom we collaborate. We will keep these definitions because, as the reviewer indicates, they are clearer to the non-specialist reader and indicate specific parameters in the work of colleagues.

Similarly. The use of the term cGDI is confusing and not helpful - this would be referred to as homing rate by most other researchers in the field.

Here again, mosquito molecular geneticists have used nomenclature that is not used broadly in the field of genetics (with apologies to some yeast geneticist). Strictly speaking, our gene-drive system achieves gene conversion via a DNA repair mechanism designated 'homology-directed repair'. We adopted 'cGDI', to acknowledge the conversions event. We added text to explain this and acknowledged the use of 'homing' by others.

It then also does not make sense to refer to them as if they were independent e.g. line 267 "The difference between the GDI and cGDI could be explained by.." - for any value in which GDI is not 100% the GDI and cGDI will always be different - they are inextricable. So on line 272 "GDI, 96.8%, and cGDI, 93.6% " are derived because $(96.8-50)/50$ is 93.6. The authors know this of course but to the uninitiated it could read as if they were independent outcomes.

We have revised the text in the Materials and Methods and Results to clarify the relationship between GDI and cGDI.

The authors describe that when molecular genotyping was carried out using Sanger sequencing that 82 F1 samples were unresolved, and labelled 'cryptic' or 'disturbed'. Given that 587 F1 genotypes were successfully accessed via Sanger sequencing, it seems a shame that the authors stopped short of investigating the remaining 82 via another approach that may yield greater resolution - such as amplicon sequencing. Indeed, the Sanger sequencing described here already makes use of PCR-based enrichment, so amplicon sequencing of these samples should be relatively straightforward. While not essential for publication, investigation of these unresolved genotypes would benefit the work and potentially reveal additional mutations.

We understand the reviewer's point. However, as he/she and Reviewer #3 acknowledge, the sequencing technique is appropriate for the testcrosses in determining germline mutagenic events. We have on-going work

looking at the complexity of genotypes in outcrosses, but as the reviewer notes it is not needed for this study. We added a brief discussion of the impact of somatic mosaicism to the Discussion.

Line 452 - change the term horizontal transfer as this could be confounded to mean horizontal transfer - I.e. between non-siblings or even between species, which is not the intended meaning of the authors

Acknowledged, this was an error and the text was changed accordingly.

typos were spotted:

- Line 154 - 'Dreamtag' PCR Mix, should presumably be 'Dreamtaq'

Corrected.

- Line 533 - 'Hammond et al. 2026', presumably 2016

Corrected.

- Line 49 - SSA is a form of homology-directed repair therefore the positioning of it as 'rather' is awry. Authors should specify HDR using the homologous chromosome as template

The text was modified to clarify this point.

Lines 512-524 the argument that "these results support a hypothesis that a threshold level of introduction must be achievedto mitigate resistance". Two things of note here: 1-the rate of accumulation per non-homing event (i.e. not per offspring) is likely the same between the two species and these observable rates allow a calculation of the propensity for resistance to arise 2- it doesn't make sense to me to contain one's thinking to the goal of suppressing a cage - at the edge of a population (or release area) there will always be areas where the drive is at low frequency, and therefore the proposed mitigation would not apply here, since resistance alleles would arise and backfill. Therefore, as written, I do not see how the data support the threshold hypothesis - at least not as a linear path to predicting an operational optimum for a strain that accumulates blocking resistance alleles in a cage setting.

In response to the first point, data from other work supports the conclusion that there are strain/species differences in the function of a specific gene-drive system (our work on the AcTP13 and AgTP13 strains of An. coluzzii and An gambaie [Carballar-Lejarazu et al., 2023]). The reviewer is correct that boundary/edge effects are likely to be important. The text was removed and the discussion will be taken up in another manuscript. .

Reviewer #3 :

While much of the manuscript is well-written, certain sections of the results section are dense and the author's decision to not express any of their findings visually (it is all tables/supplemental tables) makes it more challenging to extract key findings and concepts than it should be.

All three of the reviewers noted the density of the text in the Results and how this made it difficult to follow. We have adopted the recommendation of Reviewer #2 and added headings, subheadings and section descriptions that should make it easier.

The researchers performed extensive Sanger-based sequencing on a large number (587) of F1 individuals (presented in Table 2). The rationale for this is not made clear, as this is essentially a characterization of somatic mutations, while the purpose of the study is to characterize germline mutations that impact Cas9-based homing (data later in the study gets at this more important question).

The choice of Sanger sequencing is also quite strange, since this technology can only really give interpretable data when there is a single predominant sequence (or at most two equally proportional alleles). Somatic disruptions are almost always likely to be complex and incomplete. Thus, it is not surprising that almost all of the mosaic individuals sequenced produced no meaningful data (either no difference from WT due to frequency of somatic mutations being too low, or too many different disruptions). Really, the authors should have used a

high-throughput method (ONT or Illumina) if their objective was to actually characterize these somatic events. But again, there doesn't seem to be a clear rationale for characterizing them in the first place.

Reviewer #2 also questioned the use of Sanger sequencing. While it is conceded that it is appropriate for the revealing the genotypes of the progeny of the testcrosses, it does not fully identify the complexity of the genotypes of the outcrosses. As the Reviewer notes, this would be best done using the progeny of pair matings and a different sequencing approach. This work is in progress but is not relevant to the focus of this work, which they acknowledge is being able to distinguish inherited mutations and those generated in the germline. The purpose of the outcross progeny sequencing was to try and identify possible F1-generated mutations that could be inherited later in the testcrosses. The cryptic and disturbed traces result principally from somatic mosaicism and were not relevant to this study. We have added text to explain our reasoning.

The end result is only 53 deletion events, presented in detail in Tables/supplemental tables (using ONT or illumina on even 1 individual would probably reveal as many or more events, again reinforcing that the authors effectively produced very little data despite great effort).

A total of 957 samples were sequenced yielding 336 mutant alleles, most of which were indels. Once the cluster effects were accounted for, this resulted in the 53 to which the reviewer refers. The point was to try to find the rates at which they were generated and inherited. Yes, a lot of work, but we were looking at what in these Anopheles strains are relatively, low frequency events. Now we know.

I was surprised there was no figure that summarized these events, since that would make it much easier to observe patterns/relationships. Instead, the authors walk us through a very long description of the numbers of each type, which is difficult to follow.

Figure 1 summarizes the experimental design and Tables 1 and 2 summarize the outcomes with Tables S1-S3 presenting the primary data. We have added text to the figure legends to provide a more detailed explanation of the images. We looked at a number of options for figures, but all added additional complexity. We are not sure how and in what form a figure would make the data clearer.

Finally, we get to the real experiment, test-crossing F1 individuals ... However, as before, the authors do not synthesize the results into any figures to help the reader understand the main findings. Instead, there is a long and difficult to follow description of various classes/categories with continuous reference to supplemental tables.

Similar to the response above, Figure 2 summarizes the experimental design and Tables 3 and 4 summarize the outcomes with Tables S4-S8 presenting the primary data. Here again, we are not sure how a figure would make this any clearer.

Lines 324-326, the authors assume that "A single cluster in a replicate cross most likely represents multiple progeny inheriting the identical allele from the products of a single spermatogonial or oogonial cell in the germline of a male or female parent, respectively". However, the authors performed pooled crosses, so this approach will underestimate the rate of occurrence of common deletions and overestimate the rate of rare events. I could not find in the methods section any mention of the size of each cross, so it is impossible to determine the expected scale of this distortion (big difference between 5M x 5F and 50M x 50F, the latter would be expected to have many more common events). The assumption made by the authors would be valid only if they had performed single pair crosses (which would have been a much more powerful dataset), but they did not. This needs to be clearly stated.

This was an oversight, and we have added the information to the Materials and Methods. The point is well-taken and we avoided using numbers derived from samples that would skew the calculations (new Table 5).

16 samples contained two different mutant alleles in addition to the expected DEL11 test cross allele. This is fascinating and could be the result of fertilization with more than 1 sperm. Is it known how often this occurs in Anopheles?

We continue to ponder this result. We added the following text to the discussion: 'It is difficult to explain the recovery of testcross samples with multiple B alleles (B/B). Polyandry (one female mating with one or more genetically-distinct males) has been observed in both field and laboratory anopheline populations (Mahmood and Reisen, 1980; Tripet et al., 2003). Invoked here as a possible cause for the multiple alleles then requires polyspermy, multiple sperm entering a single oocyte. Polyspermy also has been observed microscopically in mosquitoes (reviewed by Degner and Harrington, 2016) although it is expected to lead to unstable polyploidy and aneuploidy and ultimately cell death. However, this explanation is confounded further by the recovery of all B/B genotypes in clustered samples. We are reluctant to attribute the results to experimental error considering the diligence with which the work was carried out and follow-up investigations by us and others may reveal some aspect of the biology that we are missing.'

The overall mutation rate estimates are important findings, and would be better represented in the main manuscript rather than the supplement (Table S8).

We agree and have moved it to the main text as Table 5.

Not sure why the term "tear" is used in place of mosaic. The latter term is widely understood by the genetics community and the change thus makes the manuscript harder to follow.

The point is well-taken. However, 'tear' is a phenotypic description, and we have added text in multiple places to remind the reader that it refers to mosaicism.

The discussion can be streamlined as there is a duplicative re-presentation of the results (Lines 457-474)

The text has been edited to eliminate redundancy and provide clarity. In addition, some text was moved to the Supplemental Materials.

August 14, 2025

RE: GENETICS-2025-308475

Dr. Anthony A. James
University of California, Irvine
Molecular Biology and Biochemistry
3205 McGaugh Hall
Irvine, California 92697-3900

Dear Dr. James:

Congratulations, your manuscript titled "Mutant allele formation and inheritance during Cas9/guide RNA-mediated gene drive in a population modification mosquito strain for human malaria control" is accepted for publication in GENETICS! Many thanks for submitting your research to the journal.

As part of our efforts to make titles of articles published in GENETICS more accessible to our broad readership, we often suggest different titles for accepted manuscripts. We offer these variants for your consideration though we'll use whatever title you include in the final version of your manuscript that you submit to the Editorial Office.

Title suggestions:

The impact of gene drive-resistant alleles on gene drive dynamics in mosquitos

Accounting for gene drive-resistant alleles in mosquitos with a Cas9 gene drive for malaria control

Accounting for Cas9 gene drive-resistant alleles in engineering mosquitos for malaria control

To Proceed to Publication:

1. Format your article according to GENETICS style: <https://academic.oup.com/genetics/pages/author-guidelines>
2. Ensure that you comply with data and community resource citation guidelines: <https://academic.oup.com/genetics/pages/author-guidelines#section-5-9-2>
3. Upload your final files at <https://genetics.msubmit.net>
4. Add oupsupport@scipris.com and genetics.oup@novatechset.com (or the domains @scipris.com and @novatechset.com) to your email program's "safe senders" list. You will be contacted by both at various points during the production process.

Notes:

- Your currently-accepted manuscript (unedited, as submitted, reviewed, and accepted) will be published at GENETICS and deposited into PubMed as an Advance Access article. Notify sourcefiles@thegsajournals.org before signing your license if you do not wish to publish your article via Advance Access.
- We invite you to submit an original color figure related to your paper for consideration as cover art. Please email your submission to the editorial office or upload it with your final files. You can submit a small-sized image for evaluation, and if selected, the final image must be a TIFF file 2513px wide by 3263px high (8.375 by 10.875 inches; resolution of 600ppi). Please avoid graphs and small type.
- After files are sent to Oxford University Press we use SciPris to manage article licensing and payment. If you do not have a SciPris account, you will receive an email from no-reply@scipris.com to sign up to use Oxford University Press' author portal. After logging in, follow the online instructions to sign your license and arrange any payment due.

If you have any questions or encounter any problems while uploading your accepted manuscript files, please email the editorial office at sourcefiles@thegsajournals.org.

Sincerely,

Jill Wildonger

Associate Editor
GENETICS

Approved by:
Kate O'Connor-Giles
Senior Editor
GENETICS